# Low resistance $p$-type contacts to monolayer WSe$_2$ through chlorinated solvent doping

Lauren Hoang [1], Robert K. A. Bennett[1], Anh Tuan Hoang [2], Tara Peña [1], Zhepeng Zhang[2], Marisa Hocking[2], Ashley P. Saunders [3], Marc Jaikissoon [1], Fang Liu [3], Eric Pop [1,2,4] & Andrew J. Mannix [2]✉

Tungsten diselenide (WSe$_2$) is a promising $p$-type semiconductor limited by high contact resistance ($R_C$) and the lack of a reliable doping strategy. Here, we demonstrate that exposing WSe$_2$ to chloroform provides simple and stable $p$-type doping. In monolayer WSe$_2$ transistors with Pd contacts, chloroform increases the maximum hole current by over 100× (>200 μA/μm), reduces $R_C$ to ~ 2.5 kΩ·μm, and retains an on/off ratio of 10$^{10}$ at room temperature. These improvements persist for over 8 months, survive a 150 °C thermal anneal, and remain effective down to 10 K, enabling a cryogenic $R_C$ of ~ 1 kΩ·μm. Density functional theory indicates that chloroform strongly physisorbs to WSe$_2$, inducing hole doping with minimal impact on the electronic states between the valence band and conduction band edges. Auger electron spectroscopy and atomic force microscopy suggest that chloroform intercalates at the WSe$_2$ interface with the gate oxide, contributing to doping stability and mitigating interfacial dielectric disorder, though further studies are needed to conclusively confirm this mechanism. This robust, scalable approach enables high-yield WSe$_2$ transistors with good $p$-type performance.

Two-dimensional (2D) semiconductors, particularly transition metal dichalcogenides (TMDs), are promising candidates for next-generation, high-density, complementary-metal-oxide-semiconductor (CMOS)[1,2] and low-temperature electronics. However, the large contact resistance ($R_C$) often observed in nanoscale TMD devices poses a significant obstacle to device performance, limiting the on-state drain current, $I_D$, needed for practical circuit applications. Both $n$-type and $p$-type transistors are critical for low-power CMOS[3], but progress on minimizing $R_C$ has largely been limited to $n$-type devices[4,5]. Developing scalable, low-$R_C$ $p$-type contacts for 2D transistors remains a critical challenge. Additionally, $R_C$ typically increases further at low temperatures, impeding other fundamental charge transport studies[6].

Various strategies have been explored to reduce $R_C$ to $p$-type WSe$_2$ transistors, including transferred metal[7] or semimetal[8,9] contacts. However, metal contacts typically form large Schottky barriers at the metal-2D semiconductor interface (preventing low $R_C$), and semimetal

contacts have yet to experimentally demonstrate superior performance for $p$-type devices. An alternative is to lower $R_C$ by implementing stable $p$-type doping near the contacts. Substitutional doping with electron acceptors (e.g., V[10], Nb[11,12]) is stable due to covalent chemical bonding but is likely to require complex fabrication with multiple material growth steps. In comparison, $p$-type surface charge transfer doping (SCTD)[13–24] withdraws electrons from the 2D channel using higher electronegativity capping or adsorbate layers with work function values below the Fermi-level of the WSe$_2$, such as transition metal oxides (MoO$_x$[13] and WO$_x$[14,17,18]), NO$_x$[19,20,25,26], and halide compounds (HAuCl$_4$[21], AuCl$_3$[22], RuCl$_3$[23,24], PtCl$_4$[16]). SCTD typically preserves the host lattice and has the potential to introduce fewer scattering centers[27]. However, the temporal and thermal stability of these methods remains unclear due to the high chemical reactivity or low thermal stability of the reagents involved[28]. Furthermore, there is little consensus on the mechanism of halide-based doping, with some studies

[1]Department of Electrical Engineering, Stanford University, Stanford, CA, USA. [2]Department of Materials Science and Engineering, Stanford University, Stanford, CA, USA. [3]Department of Chemistry, Stanford University, Stanford, CA, USA. [4]Department of Applied Physics, Stanford University, Stanford, CA, USA. ✉e-mail: ajmannix@stanford.edu

suggesting reactions in which Cl atoms substitute and passivate chalcogen vacancies[29–31], while others propose molecular physisorption[22,32] or intercalation[33].

Solvent exposure can also unintentionally dope TMDs. For example, $MoS_2$ and $WSe_2$ are $n$-doped by exposure to the low-electronegativity solvent acetone during removal of electron-beam (e-beam) lithography resists like poly(methyl methacrylate) (PMMA)[34,35]. Conversely, the high-electronegativity solvent chloroform ($CHCl_3$) was shown to $p$-dope semimetallic graphene[33]. This suggests that chloroform could serve as an effective $p$-type dopant for 2D semiconductors such as $WSe_2$, offering a simple and scalable approach compared to existing doping techniques. However, solvent-based doping is often regarded as transient, and the impact of chloroform doping on the electrical performance of $p$-channel $WSe_2$ transistors has not yet been studied.

In this work, we demonstrate that chloroform can induce high-performance, stable, and high-yield $p$-doping in monolayer $WSe_2$ transistors (Fig. 1a, b). Exposing monolayer $WSe_2$ transistors to chloroform increases $I_D$ by two orders of magnitude, with hole currents reaching up to 203 µA/µm at $V_{DS} = -1$ V (Fig. 1c). These devices also maintain large $I_{on}/I_{off}$ ratios (~ $10^{10}$) and a low $R_C$ of 2.5 kΩ·µm (at room temperature) and 1.0 kΩ·µm (at 10 K). Compared to recent approaches such as contact engineering (e.g., Sb/Pt[8,36], Ru[37,38]), oxide-based doping ($WO_x$[17,39], $MoO_x$[8,40], $NO_x$[19,39]), and other halide-based dopants ($HAuCl_4$[21], $RuCl_3$[23]), chloroform doping achieves one of the highest reported values for $p$-type transistor current (Fig. 1b). Additionally, we observe that chloroform-doped transistors remain stable over 8 months (retaining 81% of initial doped $I_{D,max}$) and survive annealing at 150 °C. Density functional theory (DFT) reveals that chloroform binds strongly ( > 260 meV, i.e., >10$k_B T$ at 296 K) to $WSe_2$ without introducing mid-gap states. Atomic-force microscopy (AFM), Auger electron spectroscopy (AES), and X-ray photoelectron spectroscopy (XPS) suggest that chloroform intercalates at the $WSe_2$ interface with the gate oxide, further stabilizing its doping. This straightforward approach enhances $p$-type performance in $WSe_2$ transistors and complements other contact and interface engineering techniques for advancing 2D semiconductor technologies.

## Results and discussion
### Characterization of chloroform-doped WSe₂
Optical spectroscopy provides insights into the physical and chemical interactions between the $WSe_2$ and chloroform. Raman spectra of monolayer $WSe_2$ soaked overnight in chloroform show no significant changes in the E′/A₁′ peak intensity ratio (Fig. 1d), which suggests that long-term chloroform exposure does not significantly increase the $WSe_2$ defectivity. We also do not observe an increase in LA(M) or

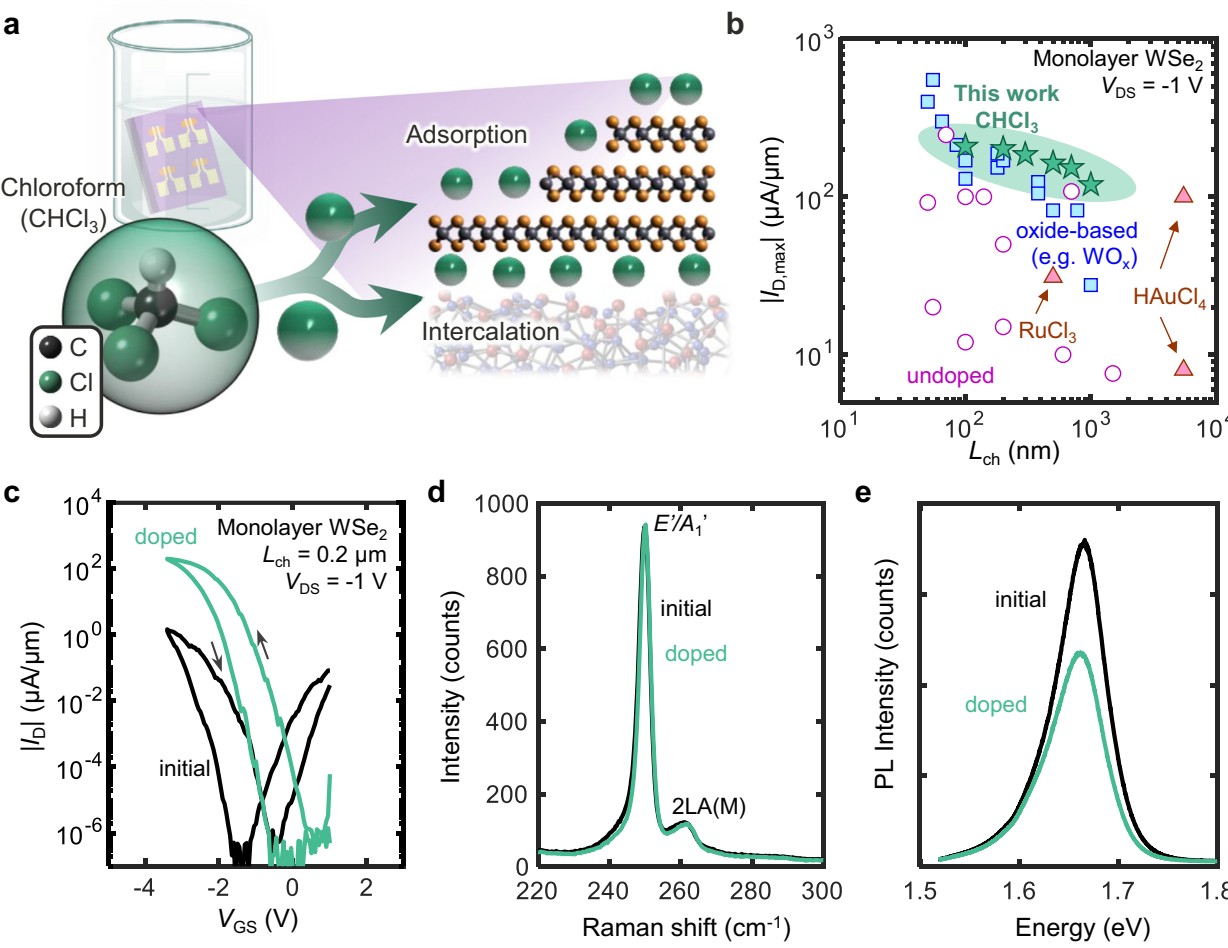

**Fig. 1 | $p$-type doping of monolayer WSe₂ using chloroform. a** Schematic of chloroform-doped $WSe_2$, illustrating the doping process and possible adsorption pathways. After fabrication, devices are left in chloroform overnight. **b** Benchmarking maximum $p$-type current ($|I_{D,max}|$) vs. monolayer $WSe_2$ channel length ($L_{ch}$) at drain voltage ($V_{DS}$) = -1 V at room temperature, using various contact metals and doping strategies. Circles[8,37–39,75–81] mark results with no intentional doping, squares[8,17,19,25,36,39,40] denote oxide-based doping ($MoO_x$, $WO_x$, $NO_x$), and triangles[21,23] label halide-based doping. Our results with chloroform doping (stars) achieve among the highest hole currents to date for monolayer $WSe_2$. **c** Measured drain current ($I_D$) vs. gate voltage ($V_{GS}$) for monolayer $WSe_2$ device before (black line) and after (green line) chloroform doping, reaching hole current of 203 µA/µm. Forward and backward sweeps are shown, revealing some counterclockwise hysteresis. **d** Raman spectra before and after chloroform doping of monolayer $WSe_2$. **e** Photoluminescence (PL) spectra of monolayer $WSe_2$ before and after chloroform doping.

2LA(M) peak intensity associated with the disruption of the $WSe_2$ lattice (Supplementary Fig. 1). The photoluminescence (PL) spectrum after chloroform doping (Fig. 1e) shows lower intensity than for the undoped sample. This PL quenching is consistent with a chloroform-induced increase in the hole concentration, leading to more non-radiative recombination via positive trions[34]. Additionally, the negligible change in surface roughness and morphology after doping indicates that residue adsorption does not play a significant role (Supplementary Fig. 2).

To investigate the electrical performance of chloroform-doped $WSe_2$, continuous monolayer CVD-grown $WSe_2$ was transferred onto an array of prefabricated ~5 nm $HfO_2$ local back gates (Fig. 2a). The local back gates were defined by photolithography and lift-off of 2/8 nm Ti/Pt, followed by thermal atomic-layer deposition of $HfO_2$ gate dielectric with equivalent oxide thickness of 1.23 nm. The $WSe_2$ channel was patterned by $XeF_2$ etching. Fine contact regions were also defined using e-beam lithography with a bilayer PMMA resist stack. Pd/Au (20/20 nm) were deposited by evaporation at ~$10^{-7}$ Torr, followed by lift-off in acetone overnight, then rinsed in isopropanol (IPA). Electrical measurements were conducted in a vacuum at ~$10^{-4}$ Torr. After initial device measurements, the devices were soaked in chloroform for >8 h and re-measured in vacuum (see Methods for more details).

Figure 2b presents the transfer characteristics of 101 transistors before and after p-doping at $V_{DS} = -1$ V, with channel lengths ($L_{ch}$) ranging from 100 nm to 1 μm. Prior to doping, the devices exhibit a highly negative $V_T$ around −2.6 V. After doping, the $V_T$ shifts positively and the maximum $I_D$ uniformly increases by ~100× across all devices, from ~1 μA/μm to >100 μA/μm (Fig. 2c). The low device-to-device variation after doping demonstrates the reproducibility of this doping method. Noticeably, the electron branch is strongly suppressed after doping (Fig. 2b). This suppression likely arises from the positive $V_T$ shift from increased hole concentration (thus requiring higher $V_{GS}$ for electron injection) and from the increase in electron Schottky barrier height, which together hinder electron current.

We can further understand the origin of these improvements by examining the effects of chloroform upon the statistical distributions of the maximum hole current $I_{D,max}$ (at $V_{GS} = -3.4$ V) and the on-state hole current $I_{on}$ at a fixed overdrive voltage ($V_{ov} = |V_{GS} - V_T|$), both shown in Fig. 2c. Interestingly, devices of all channel lengths show concurrent increases in max $I_D$, and in $I_{on}$ at fixed $V_{ov} = 1.3$ V. Evidently, the chloroform doping shifts $V_T$ positively, but the observed increase in $I_{on}$ at fixed $V_{ov}$ in both long- and short-channel devices suggests the improvement is a combined effect of increased mobility and reduced $R_C$. The maximum transconductance ($g_m$) of each device shows a 30.5× median increase (from 1.07 to 33 μS/μm) for $L_{ch} = 1$ μm devices (Supplementary Fig. 3a), consistent with this interpretation.

All devices demonstrate a positive $V_T$ shift ($V_T$ extracted at a constant current 10 nA/μm)[41], with a median shift value of 1.0 V (from −2.6 V to −1.6 V), consistent with p-doping (Fig. 2d). This $V_T$ shift corresponds to roughly $0.9-1.8 \times 10^{13}$/cm² carriers induced from this doping technique (Supplementary Fig. 3b). Precise quantification of the initial doping concentration in the $WSe_2$ is difficult as fabrication-induced effects (e.g., adsorbates, annealing, processing history, etc.) can alter the concentration. We estimate an initial electron concentration on the order of $10^{12}$ cm⁻² prior to doping (see Supplementary Note 1). Thus, upon chloroform doping, the induced hole density of ~$10^{13}$ cm⁻² can compensate this initial doping and dominate the final carrier concentration, shifting the device to p-type operation.

Additionally, unlike other SCTD methods[13,19,21], the chloroform-doped transistors did not exhibit any degradation in off-state current even at the shortest $L_{ch}$ of 100 nm. We extract the $R_C$ of our Pd-contacted doped monolayer $WSe_2$ devices using the transfer length method (TLM), yielding an $R_C$ of 2.5 (2.8) kΩ·μm for our best (median) pseudo-TLM structure (as described in Methods) (Fig. 2e). In comparison, the initial $R_C$ before chloroform exposure was 168 kΩ·μm (Supplementary Fig. 3c). The improved $R_C$ likely stems from the doping of the $WSe_2$ region near the contacts, which narrows the metal-semiconductor energy barrier width and enhances the contribution of tunneling[42,43]. We estimate a low Schottky barrier height (SBH) of ~100 meV after doping, consistent with previous theoretical calculations[44]. However, direct extraction of SBH from temperature-dependent data remains challenging due to the small screening length and large contribution of tunneling current across the narrow barrier. Rigorous simulations to extract SBH are an important future topic to understand the $R_C$ improvement in these and other doped $WSe_2$ devices. Notably, this $R_C$ value represents the best reported for Pd contacts on monolayer $WSe_2$ and is comparable to the highest-performing contact schemes reported to date (e.g., Sb/Pt with $MoO_x$ doping[36], $WO_x$ and NO doping[39]). Supplementary Table 1 benchmarks the performance of p-type monolayer $WSe_2$, highlighting that our devices achieve state-of-the-art $R_C$ and performance metrics.

Comparison of the subthreshold swing (SS) before and after doping reveals a decrease in SS for the doped devices, down to 81.4 mV/dec from 144 mV/dec at room temperature (Fig. 2f and Supplementary Fig. 3d). This improvement in SS spans the entire subthreshold range of $I_D$ from doping. (Fig. 2f and Supplementary Fig. 3e), which may be due to passivation of interfacial defects and could also explain the increase in mobility[45].

High $R_C$ also limits the operation of $WSe_2$ transistors at cryogenic temperatures, impeding the study of $WSe_2$ in quantum transport devices. Chloroform-doped monolayer $WSe_2$ transistors at 10 K (Fig. 2g) show consistently high hole current across all devices, up to 403 μA/μm at $V_{DS} = -1$ V for $L_{ch} = 0.1$ μm (Supplementary Fig. 4), with relatively linear $I_D$ vs. $V_{DS}$ (Fig. 2h). The cryogenic $R_C$ ~1.0 kΩ·μm was extracted from a pseudo-TLM fit to devices ranging from $L_{ch} = 100$ nm to 1 μm (Fig. 2i). To our knowledge, this is the lowest p-type $R_C$ reported to date for cryogenic temperatures.

## Charge transfer mechanism of chloroform-doped $WSe_2$
Temperature-dependent PL measurements of $WSe_2$ before (Fig. 3a) and after (Fig. 3b) chloroform exposure provide additional evidence for charge transfer. In both samples, the peaks at 1.75 eV and 1.71 eV correspond to neutral excitons (X) and trions (T), respectively, with the 40 meV difference matching the reported trion binding energy[46,47]. Additionally, the relative intensity of the trion peak is greater than the exciton peak in the chloroform-exposed $WSe_2$ (Supplementary Fig. 5), as expected for increased doping[48].

In both doped and undoped $WSe_2$, the three lower-energy peaks— labeled L1 (~1.67 eV), L2 (~1.64 eV), and L3 (~1.60 eV)—resemble previous reports of excitonic bound states[48] which display sublinear excitation power dependence (Supplementary Fig. 5) and rapidly quench above 100 K[49]. These characteristics are consistent with weakly-bound defect or donor states near the valence band maxima[49,50]. The intensity of the L1, L2, and L3 peaks is significantly higher in the doped sample (Fig. 3a, b), which indicates increased radiative recombination of electrons and holes bound to different sites and could be explained by the Fermi-level moving towards the valence band after chloroform doping[51,52]. The decrease in exciton intensity and emergence of bound states suggest that the passivation of $WSe_2$ defects is unlikely. Techniques[53–55] that passivate defects commonly exhibit an increase in quantum yield and the suppression of low-energy defect peaks at low temperatures.

To investigate the p-doping mechanism, we modeled the interactions between chloroform and $WSe_2$ using density functional theory (DFT). The chloroform absorption site was determined by relaxation against a rigid 5 × 5 $WSe_2$ supercell, considering geometries where the hydrogen atom faced towards (H-facing) or away from (Cl-facing) the monolayer $WSe_2$. Additional computational details are provided in the Methods section.

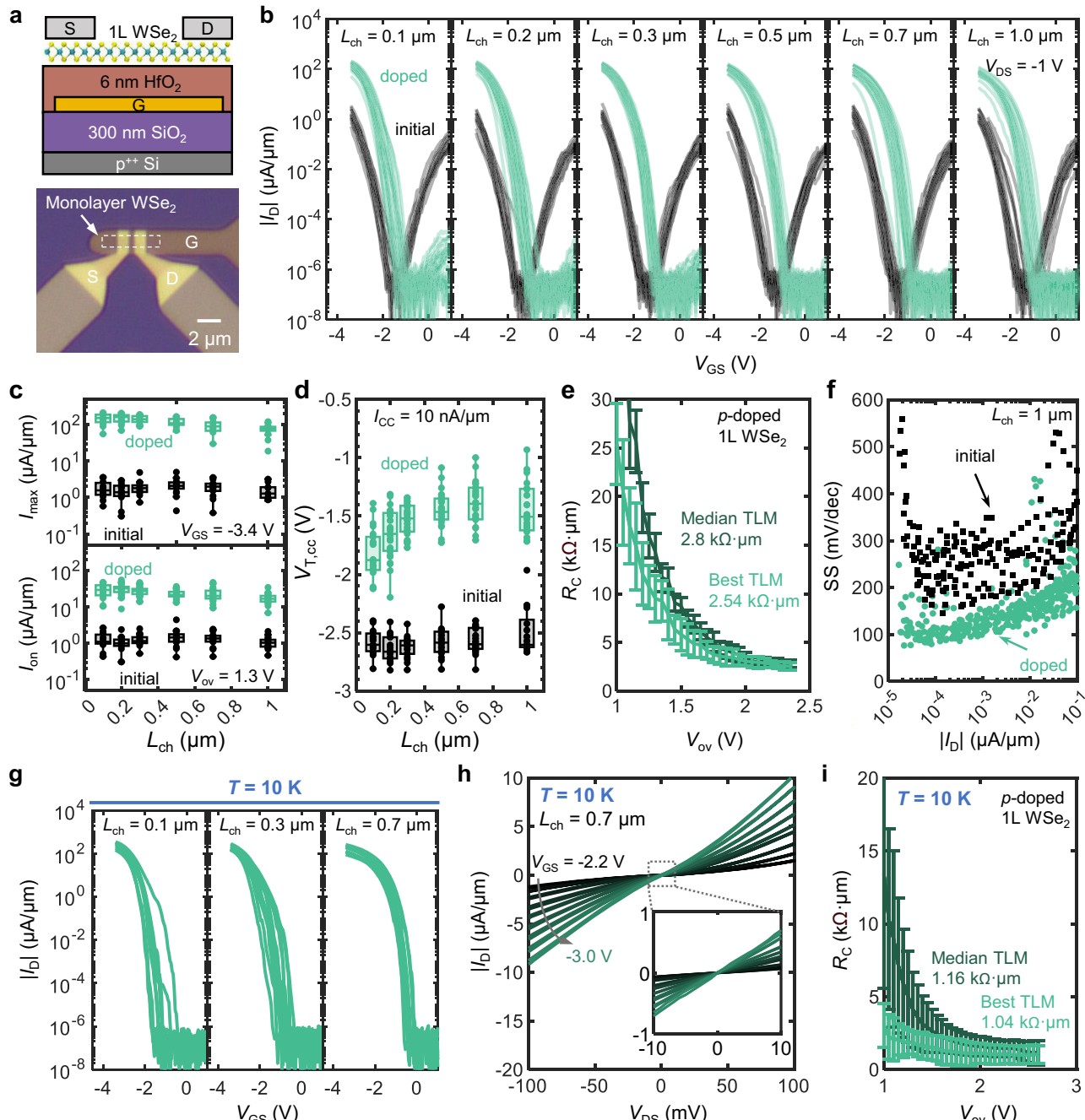

**Fig. 2 | Electrical characterization of chloroform-doped monolayer WSe₂ transistors. a** Cross-sectional schematic of the WSe₂ transistor (top) and optical microscope image of the fabricated device (bottom). **b** Measured $I_D$ vs. $V_{GS}$ before and after doping at several channel lengths ($L_{ch}$) from 0.1 to 1 μm. **c** $L_{ch}$-dependent statistical analysis before and after doping of (top) maximum drain-current $I_{D,max}$ at $V_{GS}$ = −3.4 V, and (bottom) on-state current $I_{on}$ at an overdrive voltage $V_{ov}$ = 1.3 V. **d** Threshold voltage ($V_{T,cc}$) at a constant current of 10 nA/μm before and after doping. All devices display a positive shift in $V_T$, indicating *p*-doping. In panels (**c**, **d**), a box plot was created for every group of data. The central mark of the box indicates the median, and the bottom and top edges of the box indicate the 25th and 75th percentiles, respectively. **e** Contact resistance ($R_C$) of chloroform-doped WSe₂ devices with Pd contacts, extracted using the transfer length method (TLM). **f** Subthreshold swing (SS) vs. $I_D$ in $L_{ch}$ = 1 μm devices before and after doping. Doped devices show lower SS across the whole subthreshold $I_D$ range. **g** Measured $I_D$ vs. $V_{GS}$ at 10 K after doping for various channel lengths ($L_{ch}$ = 0.1 to 0.7 μm). **h** $I_D$ vs. $V_{DS}$ curve for a representative $L_{ch}$ = 0.7 μm device at 10 K from $V_{GS}$ = −3.0 V to −2.2 V in steps of 0.1 V increments. The inset shows a magnified view of the low-voltage region. **i** $R_C$ of chloroform-doped WSe₂ devices with Pd contacts at 10 K, extracted using the TLM method. Notably, a low contact resistance and high drain current are still maintained at cryogenic temperatures. In panels (**e**, **i**), error bars indicate the standard error of the linear fit used for TLM extraction (fitting total resistance vs. $L_{ch}$ at fixed $V_{ov}$), representing the uncertainty in the extracted $R_C$.

The Cl-facing and H-facing chloroform orientations exhibit favorable adsorption energies ($E_{ads}$) of −468 and −351 meV, respectively, indicating strong physisorption ($|E_{ads}| \gg k_B T$) to the WSe₂ without inducing covalent chemical modification (Fig. 3c, d). This greatly exceeds adsorption energies between some small molecules and graphene ($|E_{ads}| < 100$ meV)[56] and are on the high end of values calculated for other adsorbates on TMDs (50 to 333 meV)[57,58]. In comparison, chemisorption is characterized by stronger binding energies (≥1 eV) and shorter distances (<2 Å)[59]. Thus, the adsorption energies and equilibrium distance from 2.5–3.2 Å (Fig. 3d) resemble strong physisorption.

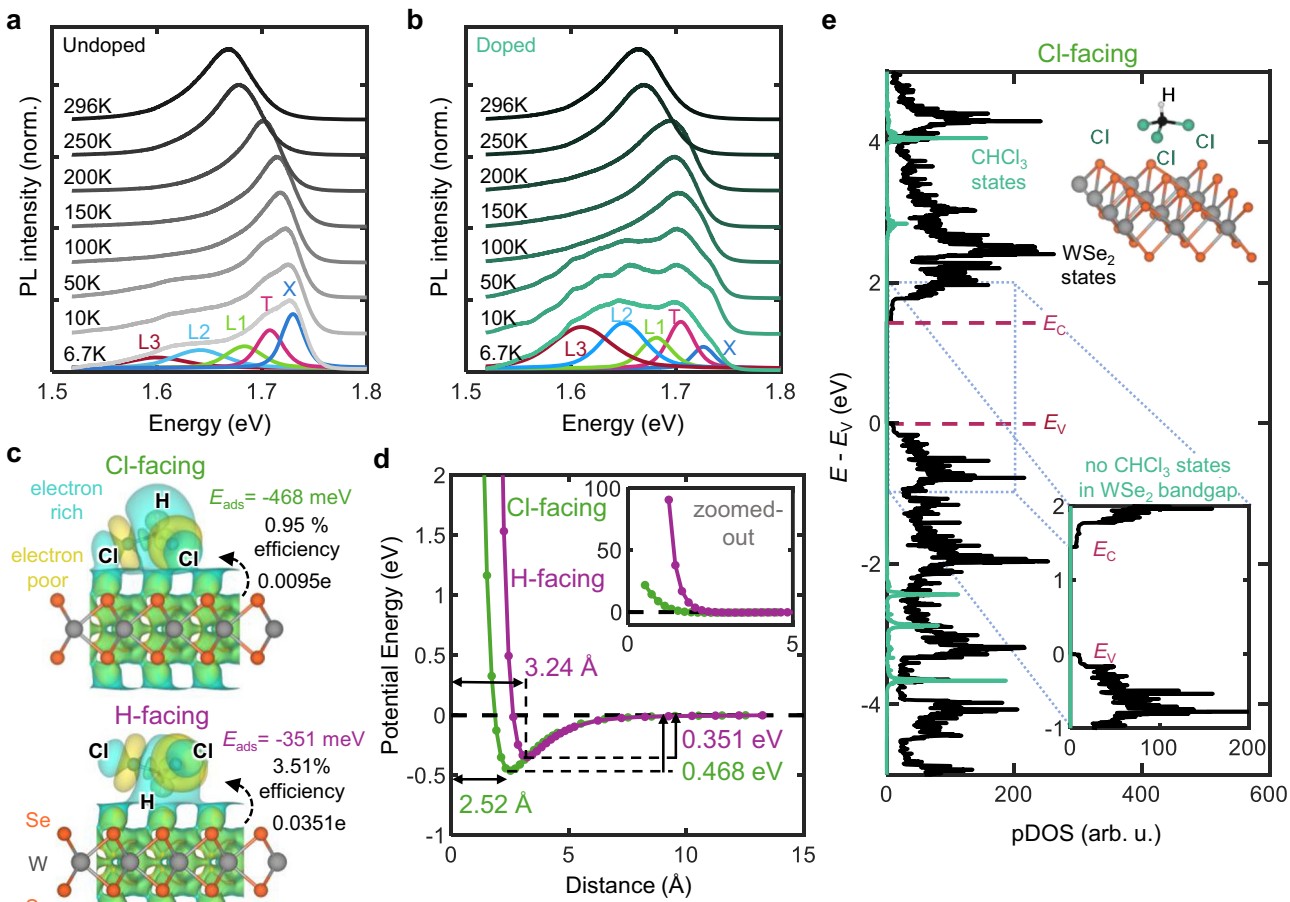

**Fig. 3 | Charge transfer mechanism of monolayer WSe₂ doped with chloroform. a** Photoluminescence (PL) spectra of an undoped monolayer WSe₂ sample at different temperatures (6.7 to 296 K). **b** PL spectra of chloroform-doped monolayer WSe₂ sample from 6.7 to 296 K. Representative Gaussian-Lorentzian blend curve fits are shown for the 6.7 K spectra in panels (**a**, **b**), corresponding to the neutral exciton (X), trion (T), and L1-L3 peaks (described further in the text). **c** Density functional theory (DFT) simulated isosurfaces of monolayer WSe₂ with adsorbed chloroform in Cl-facing (top) and H-facing (bottom) configurations, as well as the calculated adsorption energies ($E_{ads}$). The value of the Bader charge transfer efficiency is shown for each chloroform orientation, corresponding to

the charge transfer of one chloroform molecule. **d** Potential energy vs. distance for an adsorbed chloroform to monolayer WSe₂ in Cl-facing and H-facing orientations. The adsorption energy ranges from 300–500 meV and the equilibrium distance from 2–4 Å, consistent with strong physisorption. Inset: zoomed-out view of the potential energy vs. distance. **e** Projected density of states (pDOS) contributions from monolayer WSe₂ and chloroform to the overall DOS in the Cl-facing orientation. The valence band maximum $E_V$ and conduction band minimum $E_C$ are marked with dashed pink lines. Noticeably, no chloroform states are formed in the WSe₂ band gap. The inset shows a zoomed-in view of the PDOS contributions around the WSe₂ band gap.

Bader charge analysis reveals that the chloroform withdraws electrons from the WSe₂ in both configurations (Fig. 3c). In both the Cl-facing and H-facing orientations, the adsorbed chloroform molecule gains a net charge of 0.0095 and 0.0351 excess electrons, respectively, confirming *p*-doping. These electron transfers are comparable to that between several well-known TMD SCTD systems, including: (i) MoO₃, a well-established *p*-dopant, and MoS₂ (-0.077 electrons[60] transferred from MoS₂ to MoO₃ per unit cell of MoS₂, assuming full surface coverage), (ii) MoS₂ and acetone (~ 0.039 electrons transferred to MoS₂ per molecule of acetone[61]), which is known to strongly *n*-dope 2D TMDs[34,35], and (iii) nitric oxide (NO) and WS₂ (0.018 electrons per molecule of NO[57]), which is known to be an excellent *p*-dopant for 2D TMDs[19,39]. For a threshold voltage shift of 1 V, the calculated Bader charge transfer of 0.0095 (0.0351) holes per chloroform molecule corresponds to a surface coverage of 1.0 (0.27) chloroform molecules per WSe₂ unit cell. The calculated Bader charge transfer could be increased by up to a factor of four due to substrate interactions[61], which would further enhance the predicted efficiency of hole doping due to chloroform adsorption.

According to the atom-resolved projected density of states (pDOS), the chloroform orbitals are located more than 1 eV below the valence band edge or above the conduction band edge of monolayer WSe₂, and the chloroform molecule does not introduce electronic states near the band extrema or in the band gap (Fig. 3e and Supplementary Fig. 6). This suggests that the charge transfer between chloroform and WSe₂ occurs without covalent bond formation or orbital hybridization, which is consistent with the adsorption energies we calculate for the chloroform/WSe₂ system (additional states associated with chemisorption are typically accompanied by an adsorption energy < −500 meV[62]). The absence of states formed in or near the band gap suggests that chloroform doping avoids introducing scattering sites that could degrade the mobility of WSe₂.

**Temporal and thermal stability of chloroform-doped WSe₂**
To evaluate the long-term stability of chloroform doping, we regularly measured the charge transport characteristics of doped monolayer WSe₂ transistors for more than 8 months. Figure 4a shows the evolution of $I_D$ vs. $V_{GS}$ sweeps for a single doped monolayer WSe₂ device

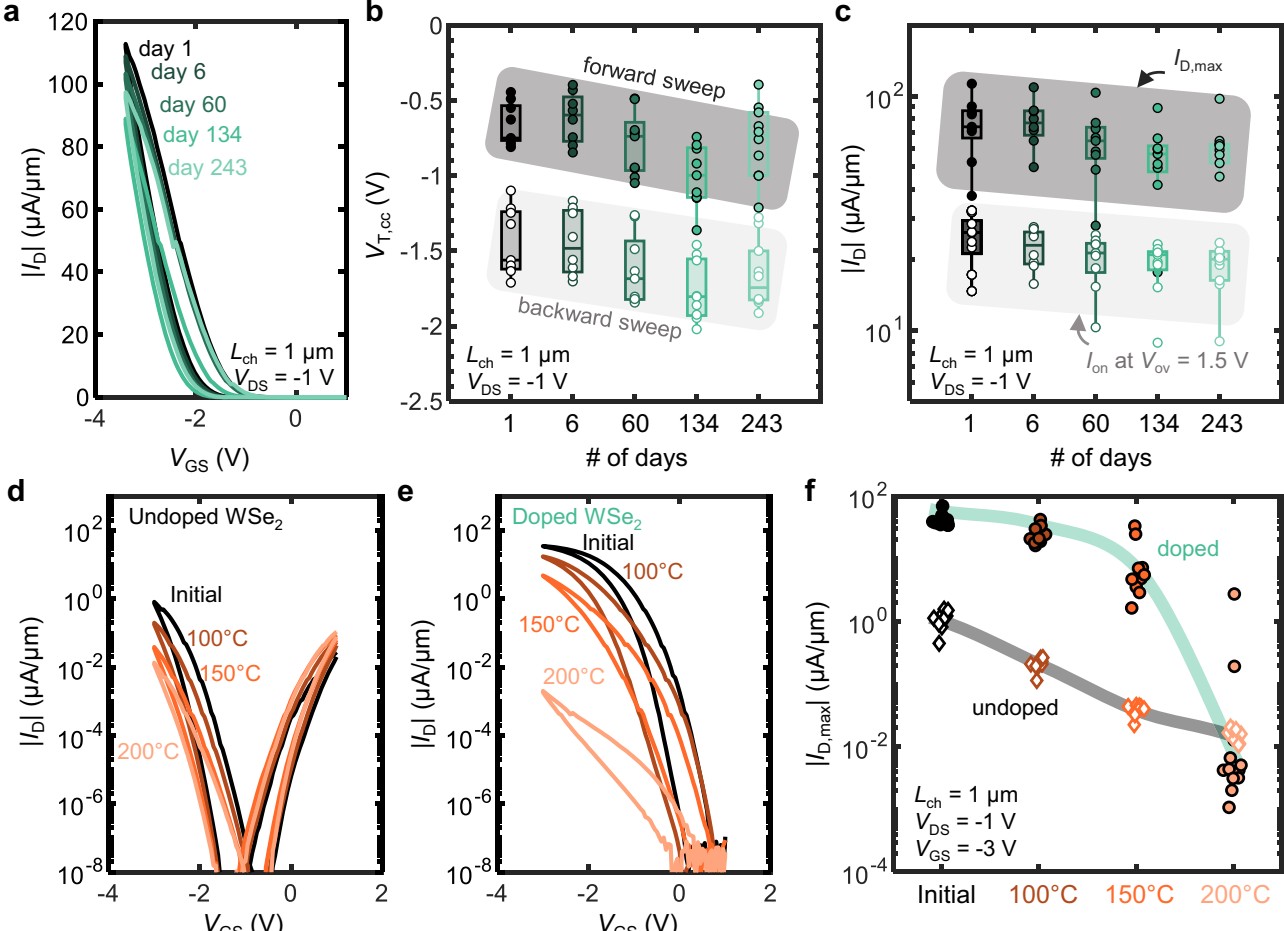

**Fig. 4 | Time and temperature stability of chloroform-doped WSe₂ devices.**
**a** $I_D$ vs. $V_{GS}$ curves of a $L_{ch} = 1\,\mu m$ device immediately after doping and after 6, 60, 134, and 243 days. **b** Threshold voltage ($V_{T,cc}$) vs. days after doping for all $L_{ch} = 1\,\mu m$ devices. $V_{T,cc}$ is extracted at a constant current of 10 nA/μm at $V_{DS} = -1\,V$ for both forward and backward sweeps. **c** Drain current (both $I_{D,max}$ at $V_{GS} = -3.4\,V$ and $I_{on}$ at $V_{ov} = 1.5\,V$) vs. days post-doping for all $L_{ch} = 1\,\mu m$ devices. For panels (**b**, **c**), a box plot was created for every group of data. The central mark of the box indicates the median, and the bottom and top edges of the box indicate the 25th and 75th percentiles, respectively. **d** $I_D$ vs. $V_{GS}$ curves of an undoped WSe₂ device ($L_{ch} = 1\,\mu m$) at $V_{DS} = -1\,V$ initially, then after annealing at 100, 150, and 200 °C. **e** $I_D$ vs. $V_{GS}$ curves of a doped WSe₂ device ($L_{ch} = 1\,\mu m$) at $V_{DS} = -1\,V$ initially, then after annealing at 100, 150 and 200 °C. **f** $I_{D,max}$ at $V_{GS} = -3.0\,V$ after various annealing temperatures, for undoped (unfilled diamonds) and doped (filled circles) devices. For the annealing process, the devices are sequentially annealed in vacuum at -10⁻⁴ Torr for 30 min at the given temperature. After annealing, the devices are cooled to room temperature for electrical measurement, then re-annealed at the next temperature.

over 243 days (>8 months). The maximum drain current $I_{D,max}$ at $V_{GS} = -3.4\,V$, slightly decreases after 243 days from 113 μA/μm to 97.4 μA/μm. Figure 4b plots the forward and backward sweep $V_{T,cc}$ for devices with $L_{ch} = 1\,\mu m$, revealing a median negative shift of −0.18 V over the course of long-term testing. This small negative shift in $V_T$ could indicate a slight reduction in p-doping due to chloroform desorption.

Figure 4c summarizes the evolution of $I_D$ in 1 μm long devices as $I_{D,max}$ for $V_{GS} = -3.4\,V$ and at $V_{ov} = 1.5\,V$. Chloroform doping remains remarkably stable over time, with the median $I_{D,max}$ retaining >96% after 6 days and >81% after 243 days. We note that the decrease in $I_{D,max}$ can be partially attributed to the negative $V_T$ shift because the $I_{on}$ remains relatively stable. After 8 months, the final $I_{D,max}$ is still 76.1× higher than the initial undoped $I_{D,max}$ for the same set of devices. This demonstrates that the improved p-type performance from chloroform doping is highly stable over time. In contrast, other doping techniques (e.g. MoO₃[13,63], O₃ oxidation[64]) degrade rapidly in air, losing functionality over the course of several hours or days. The next-best reported example, nitric oxide, maintained performance after 24 days[25]. Additionally, the low $R_C$ from chloroform doping was maintained after 8 months (Supplementary Fig. 7). This stability of chloroform doping over time is consistent with the strong physisorption predicted from our DFT simulations (Fig. 3c, d).

Thermal stability is also critical to enable further processing. We examined the thermal stability of undoped and chloroform-doped WSe₂ transistors by sequentially annealing them in vacuum for 30 min at 100 °C, 150 °C, and 200 °C. Figure 4d shows the $I_D$ vs. $V_{GS}$ of a control device, displaying a progressive decrease in hole current after each annealing step. This decline may result from the desorption of weakly-bound water molecules, which also contribute to p-doping[65,66]. Figure 4e plots the $I_D$ vs. $V_{GS}$ evolutions for a chloroform-doped device after the same annealing sequence. Similar to the undoped control device, the doped device exhibited a slight reduction in hole current after annealing at 100 °C and 150 °C. However, after the 200 °C anneal, the $I_{D,max}$ of the doped device dropped significantly to $3 \times 10^{-3}$ μA/μm, comparable to the control device under similar annealing conditions. This suggests that chloroform desorbs at elevated temperatures, reverting the device to an undoped state. This is supported by activation energy calculations (Fig. 3d), which show that there is no additional barrier to desorption, leading to much faster desorption at higher temperatures.

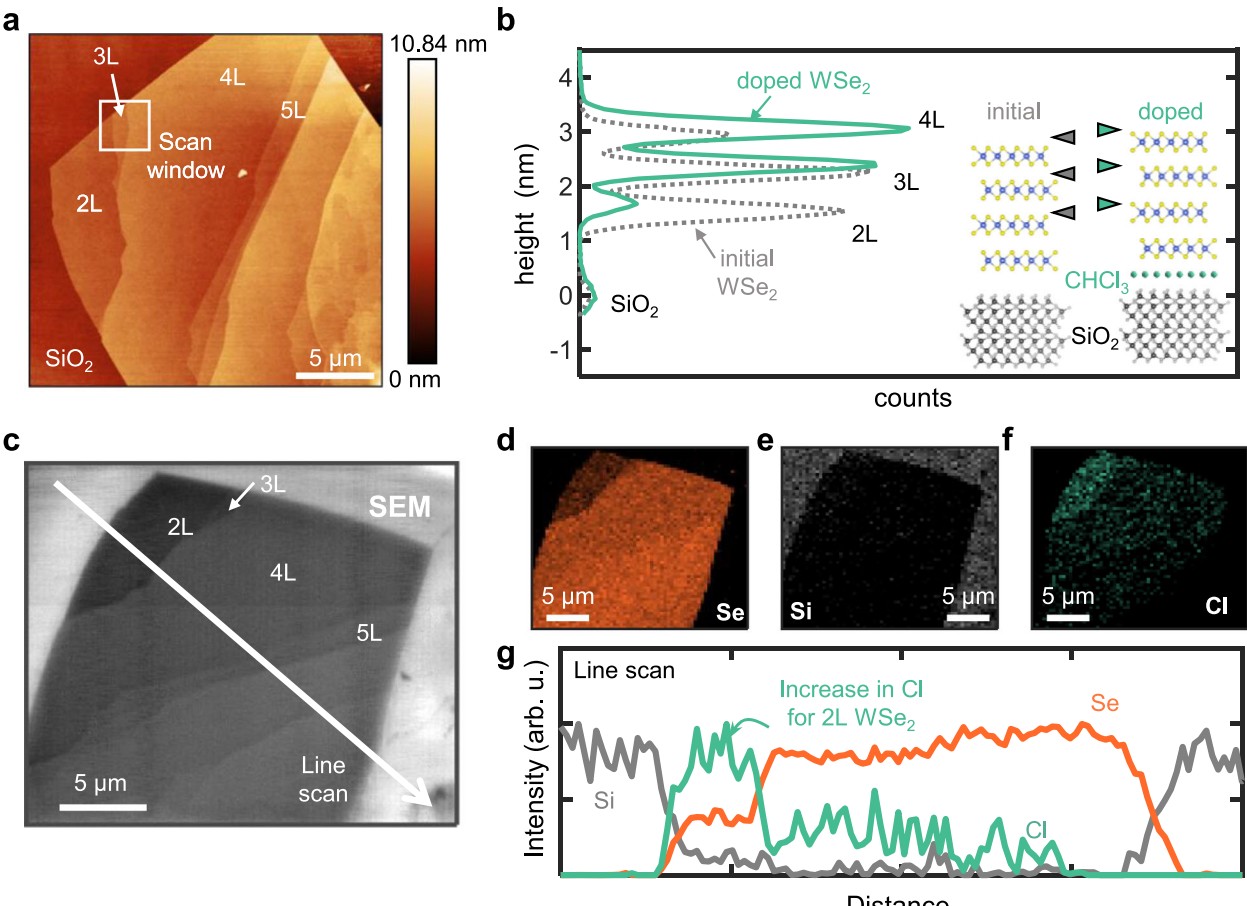

**Fig. 5 | Determination of chloroform location in a WSe₂/oxide stack. a** Atomic-force microscopy (AFM) of an exfoliated WSe₂ flake with various layer thicknesses. L denotes the number of WSe₂ layers. **b** Height distribution of the exfoliated flake before (gray) and after (green) doping, in the 2–4L region (as marked in panel (**a**)). The peaks mark the height of the SiO₂, 2L, 3L, and 4L WSe₂ regions. There is no noticeable change in spacing between WSe₂ layers, but the difference between SiO₂ and 2L WSe₂ increases. Inset: schematic of chloroform inserting at the SiO₂/WSe₂ interface, causing an increase in height of the WSe₂ layer relative to SiO₂. Triangles denote the height of the 2L, 3L, and 4L WSe₂ regions, matching the peaks in the height distribution. **c** Scanning electron microscope (SEM) image of the exfoliated WSe₂ flake as seen in panel (**a**). **d**–**f** Elemental mapping by Auger electron spectroscopy (AES) of a doped WSe₂ flake of Se, Si, and Cl, respectively. The brighter pixels correspond to regions with higher elemental content. **g** Line scan of elemental Se, Si, and Cl content extracted from panel (**d**), showing an increase in Cl signal in the 2L WSe₂ region.

A summary of $I_{D,max}$ across all annealing stages (i.e., the initial state and anneals at 100 °C, 150 °C, and 200 °C) shows that the doped devices remain >140× higher in $I_{D,max}$ after annealing at 100 °C and 150 °C (Fig. 4f and Supplementary Fig. 8). However, following the 200 °C anneal, there was a sharp drop in hole current of the doped devices, consistent with the desorption of chloroform and a reversal to the undoped state. This sequential annealing procedure suggests that 150 °C can be treated as a safe upper-bound for the thermal stability of chloroform doping on WSe₂, although faster thermal ramping and cooling may reveal a thermal budget for higher temperatures. While this thermal budget could be a concern for direct-current (DC) operation with significant self-heating[67,68], devices operating under high frequencies will heat up less. This is because the device switching speed is higher than the 2D device thermal time constant (typically from 30 to 300 ns[69]). In any case, this 150 °C thermal budget for chloroform stability enables compatibility with oxide encapsulation by atomic-layer deposition (ALD), which often occurs between 100 and 200 °C. This may also further enhance the thermal stability of the *p*-type doping, enable fabrication of top-gated devices, and allow for concurrent application of other doping techniques, such as solid charge transfer layers (e.g., MoOₓ, WOₓ).

## Determination of chloroform location in a WSe₂/oxide stack

To clarify the mechanism and stability of chloroform doping, we investigated its location relative to the WSe₂. Figure 5a presents a 20 × 20 μm² atomic-force microscopy (AFM) topography image of an exfoliated WSe₂ flake in which the thickness increases from 2 to over 10 layers. The bilayer (2L) to four-layer (4L) region was measured in 2 × 2 μm² scans before and after doping (Supplementary Fig. 9), yielding the height distributions shown in Fig. 5b. The peaks mark the height of the SiO₂, 2L, 3L, and 4L WSe₂ regions. There is no noticeable change in spacing between the WSe₂ layers, but the height difference between the SiO₂ and 2L WSe₂ increases by >0.15 nm. This suggests that chloroform does not intercalate between the WSe₂ layers, but rather that chloroform either (i) inserts at the SiO₂/WSe₂ interface, or (ii) adsorbs on top of every WSe₂ layer. X-ray diffraction (XRD) reveals that the interplanar spacing remained constant at 0.645 nm after doping (Supplementary Fig. 10), supporting the conclusion that chloroform does not intercalate between WSe₂ layers.

We measured the dependence of chloroform adsorption on WSe₂ thickness using Auger electron spectroscopy (AES). Figure 5c shows a scanning electron microscopy (SEM) image of the exfoliated WSe₂ flake from Fig. 5a, while Fig. 5d–f display AES elemental maps of Se, Si, and Cl. Figure 5g plots the AES signal intensities along the line in Fig. 5c. As a surface sensitive technique with an Auger electron escape depth of approximately 5 to 50 Å, AES confirms that the Se signal intensity scales with the WSe₂ thickness (Fig. 5d). In contrast, the Cl signal is negligible outside the WSe₂ region, peaks within the

2L WSe$_2$ terrace, and diminishes significantly for thicker WSe$_2$ layers (Fig. 5f, g).

AFM height mapping indicates uniform increases in height across the 2L, 3L, and 4L WSe$_2$ regions after chloroform exposure (Fig. 5b), while AES mapping shows the highest Cl signal in the (thinnest) 2L region (Fig. 5f, g). This suggests that chloroform intercalates at the WSe$_2$/oxide interface, as reported for graphene on SiO$_2$[33], with thicker WSe$_2$ regions attenuating the AES signals from Cl beneath the WSe$_2$. This interfacial chloroform may enhance the WSe$_2$ device performance by increasing the oxide/WSe$_2$ separation and reducing the influence of interfacial oxide $n$-doping[70] and trap states—which may contribute to the observed reduction in SS for doped WSe$_2$ devices (Fig. 2f). Although this data set is convincing, further experimental work is required to definitively confirm the intercalation of chloroform. Notably, the correlation between the Cl signal and WSe$_2$ regions suggests that WSe$_2$ is necessary for chloroform adsorption (Fig. 5f). XPS is consistent with this observation, detecting a Cl peak only in substrate regions covered by monolayer WSe$_2$ (Supplementary Fig. 11). In contrast, there is no apparent Cl peak in the bare substrate regions of chloroform-soaked samples.

We further assess the substrate dependence of our doping strategy and its implications for CMOS compatibility. While chloroform doping demonstrates reproducible $p$-type doping without leaving metallic residues on the surface, the sensitivity to standard wet-processing solvents (e.g., acetone) poses integration challenges (Supplementary Fig. 12a). Additionally, selective-area doping requires the development of protective strategies, such as encapsulation, that prevent dopant desorption (Supplementary Fig. 12b, c). Further experimental work is needed to optimize encapsulation strategies for localized doping. To further clarify the influence of the substrate, we extend our experiments to include WSe$_2$ devices on SiO$_2$ (Supplementary Fig. 13). These show comparable performance enhancement as those on HfO$_2$ (~100× increase of $I_D$, positive $V_T$ shift, and similar change in carrier concentration). This suggests that the doping mechanism is robust across different amorphous dielectrics.

Overall, this work presents a straightforward and stable $p$-doping method to achieve high-performance monolayer WSe$_2$ transistors, while providing new mechanistic insights into solvent-based doping techniques. By achieving significant improvements in hole current, $R_C$, and device stability, this method offers a viable path for future low-power 2D semiconductor applications.

## Methods

### Doping process
The WSe$_2$ sample was immersed in as-purchased undiluted chloroform (SIGMA-Aldrich, No. 650498), in a watchglass-covered borosilicate beaker at standard room temperature (20-24 °C) and relative humidity (approx. 25%-65%) in a ventilated fume hood. The chloroform solvent was used without modification (as-purchased ≥99.9% chloroform). Unless indicated otherwise, the doping process occurred overnight (>8 h). Samples were exposed only to pure chloroform, as dilution with other solvents (e.g., acetone, IPA) could introduce co-dopant effects, complicating the interpretation of concentration-dependent trends. For device measurements, the doping process was performed after the initial device fabrication process was completed.

### Material characterization
Raman measurements were taken on the Horiba Labram HR Evolution Raman system in the Stanford Nanofabrication Shared Facility, using 532 nm laser excitation at 1% nominal laser power (120 μW) and a spot size <1 μm in diameter. These parameters were selected to ensure minimal sample heating during measurement. For Raman and PL, a solid-source chemical vapor deposition (CVD) monolayer of WSe$_2$ grown on sapphire was transferred onto 100 nm SiO$_2$ before measurement. XPS was carried out using a PHI VersaProbe 4, equipped with a monochromatized Al Kα source (1486 eV) with a beam power of 50 W and beam energy of 15 kV, base pressure of $1.2 \times 10^{-7}$ Pa, and pass energy of 224 eV (step size: 0.8 eV) and 55 eV (step size: 0.1 eV) for survey and high-resolution acquisitions, respectively.

Bulk WSe$_2$ crystals were exfoliated with scotch tape onto oxygen-plasma cleaned silicon wafers with 100 nm thermal oxide. The exfoliated WSe$_2$ was probed for Auger electron spectroscopy (AES), X-ray diffraction (XRD), and atomic-force microscopy (AFM) images in Fig. 5a and Supplementary Fig. 9. AES mapping, composition analysis, and line scans on exfoliated WSe$_2$ were performed on a PHI 700 Scanning Auger Nanoprobe. XRD measurements were conducted using a PANalytic Empyrean system with a Cu-Kα source. Exfoliated WSe$_2$ flakes were probed with symmetric 2θ/ω scans. AFM was conducted on both the exfoliated WSe$_2$ and on CVD-grown WSe$_2$ on sapphire using a Bruker Dimension Icon in peak force mode with an NSC19 Al BS probe (nominal spring constant = 0.5 N/m).

### Local back-gate device fabrication on HfO$_2$ and electrical measurements
Continuous 2-inch CVD-grown monolayer WSe$_2$ on sapphire was purchased from 2D semiconductors and transferred onto local back gates of 5.3 nm HfO$_2$ with $C_{ox}$ = 2.8 μF/cm$^2$. The local back gates were defined by lift-off 2 nm/8 nm Ti/Pt followed by the HfO$_2$ gate dielectric by thermal atomic-layer deposition at 200 °C. Coarse contact pads were then defined by lift-off 2/20 nm Ti/Pt. Polystyrene (PS) was spin-coated on top of the WSe$_2$ and then transferred in DI water. An O$_2$ plasma treatment (100 W, 1 min) of the HfO$_2$ dielectric was done before transferring the PS/WSe$_2$ film to modify the substrate's surface energy. The PS was then removed in toluene. Channel definition was done using electron-beam lithography and etched by XeF$_2$ (2.5 T, 30 s, 3 cycles) to define a channel width of 1 μm. Electron-beam lithography was used to pattern the fine contacts. Pd/Au (20/20 nm) was e-beam evaporated at ~10$^{-8}$ Torr. Electrical measurements were performed at 296 K in a Janis ST-100 vacuum probe station at ~10$^{-4}$ Torr, using a Keithley 4200 semiconductor parameter analyzer.

Cryogenic measurements were conducted in a Lakeshore cryoprobe station at ~10$^{-6}$ Torr, using a Keithley 4200 semiconductor parameter analyzer. The sample was slowly cooled and left to stabilize overnight at 10 K before electrical testing.

For contact resistance ($R_C$) extraction, a pseudo-transfer length method (TLM) was used, as devices made were single devices with varying channel lengths. In this method, all devices at a certain channel length were used for $R_C$ extraction. The total resistance in kΩ·μm (normalized by the channel width) can be expressed as $R_{TOT} = 2R_C + R_{ch} = 2R_C + R_{sh}L_{ch}$, where $R_{sh}$ is the sheet resistance of the channel and $R_{ch}$ is the channel resistance. $R_C$ is evaluated by plotting $R_{TOT}$ versus $L_{ch}$ and drawing a linear fit through all data points, and the $y$-intercept at $L_{ch} = 0$ gives $2R_C$. The $R_C$ is extracted for each gate overdrive $V_{ov} = |V_{GS} - V_T|$, with $V_T$ from the constant-current method at $I_D = 10^{-2}$ μA/μm.

### Low temperature photoluminescence
Low-temperature photoluminescence spectroscopy was conducted with a 532 nm excitation laser, ~1 μm spot size, and 600 l/mm spectrometer grating. The laser power was fixed at 60 μW, unless otherwise noted. The emission was collected using a 50× objective with a numerical aperture of 0.55, with 2 s acquisition times and 2 accumulations. The sample was cooled to a base temperature of ~6.7 K, then warmed up using a resistive heater for temperature-dependent measurements. For temperature-dependent experiments, the sample sat for 30 min at the desired temperature to stabilize before collecting the spectra. For this experiment, CVD-grown WSe$_2$ was wet transferred (as described above) onto 100 nm SiO$_2$/p$^{++}$ Si, then half of

the chip was cleaved and subjected to an overnight chloroform soak. Several spots across both the control and doped samples were examined to ensure peak shape consistency. Finally, peak fitting was conducted in Origin using a Gaussian-Lorentzian blend.

### Temporal and thermal stability testing

For time stability measurements, devices were stored at room temperature in a nitrogen dry box under continuous $N_2$ purge, with relative humidity maintained at ~5%. For thermal stability testing, initial electrical measurements were performed at 296 K in a Janis ST-100 vacuum probe station at ~$10^{-4}$ Torr, using a Keithley 4200 semiconductor parameter analyzer. The samples were then in-situ annealed at 100 °C, held for 30 min, then left to cool down for >5 h. Electrical measurements were taken at 300 K in a vacuum. This process was then repeated at 150 °C and 200 °C, respectively, with device measurements in between, without breaking the vacuum. At present, the impact of ambient humidity and oxygen on the long-term stability of chloroform doping remains an open question and warrants future investigation, particularly in the context of encapsulated or integrated device architectures.

### Density functional theory (DFT) simulations

First, a variable cell relaxation was performed to optimize the lattice coordinates within the monolayer $WSe_2$ primitive cell. The optimized primitive cell was then scaled to a $5 \times 5$ supercell, interfaced with a chloroform molecule, and then subjected to a fixed cell relaxation to determine the $WSe_2$/chloroform atomic coordinates. Both the Cl-facing and H-facing orientations were considered, where the chlorine or hydrogen atom of the chloroform molecule was oriented towards the $WSe_2$. Quantum ESPRESSO 7.1[71] was used for all DFT simulations, and the van der Waals interactions between the chloroform and $WSe_2$ monolayer were modeled using the vdW-DF-C09 correction. For all self-consistent calculations, we use Γ-point sampling for all $5 \times 5$ supercells, and $k$-point grids of $3 \times 3 \times 1$ and $7 \times 7 \times 1$ for the $3 \times 3$ and $2 \times 2$ supercells, respectively. We perform non-self-consistent calculations on a $15 \times 15 \times 1$ $k$-point grid prior to extracting the density of states for the $WSe_2$ + chloroform assemblies. All DFT calculations use projector-augmented wave pseudopotentials with kinetic energy cutoffs and charge density cutoffs of 50 and 400 Ry, respectively. We use the "Bader" code[72] for Bader charge analysis, PyProcar[73] for plotting projected density of states and band structures, and VESTA[74] for plotting isosurfaces. We note that in this work, we consider only interactions between the chloroform and $WSe_2$ without an insulator or substrate. Including the amorphous $SiO_2$ and $HfO_2$ would improve the accuracy of our simulations by capturing chloroform-substrate interactions; however, simulating this interface is computationally demanding, and we leave it as a promising research direction.

Other orientations of chloroform on $WSe_2$ are certainly possible, although a rigorous test for the most favorable orientation remains outside of the scope of this study. We confirmed that the Cl-facing orientation is stable compared to other similar random orientations by rotating the chloroform molecule by 5 degrees (i.e., tilting the C-H bond 5 degrees relative to the surface normal) and then re-relaxing the system. The chloroform molecule returned close to its original position and total energy (to within 2 meV), suggesting that this orientation is indeed favorable.

### Data availability

Relevant data supporting the key findings of this study are available within the article and the Supplementary Information file. All raw data generated during the current study are available from the corresponding authors upon request.

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

## Acknowledgements

This work was supported by the National Science Foundation (NSF) FuSe2 grant 2425218 and the TSMC-Stanford SystemX Alliance. L.H. was supported by the Sandia Microelectronics: Accelerating Research Talent (SMART) Internship. L.H., R.K.A.B., T.P., and E.P. acknowledge support from SUPREME, a JUMP 2.0 center within the Semiconductor Research Corporation (SRC), sponsored by the Defense Advanced Research Projects Agency (DARPA). R.K.A.B. was supported by the Stanford Graduate Fellowship and the NSERC PGS-D programs. T.P. acknowledges support from the NSF MPS-Ascend postdoctoral fellowship. A.P.S. and F.L. were supported by the U.S. Department of Energy, Office of Science, Basic Energy Sciences, CPIMS Program, under award no. DE-SC0026181. Part of this work was performed at the Stanford Nanofabrication Facility (SNF) and Stanford Nano Shared Facilities (SNSF), supported by NSF award ECCS-2026822. The authors thank Krishna C. Saraswat, Johnathan Georgaras, Qingrui Cao, Akash Ramdas, Emily Chen, and Alex Shearer for valuable discussions.

## Author contributions

L.H. fabricated the devices and conducted the device measurements and analysis under the supervision of A.J.M. and E.P. L.H. conducted the AFM, Raman, and Auger measurements. R.K.A.B. performed the DFT simulations. A.T.H. performed the CVD $WSe_2$ material growth. T.P. and L.H. performed the low-temperature PL characterization and analysis with the help of A.P.S. under the supervision of F.L. Z.Z. performed the XRD with L.H. M.H. performed the $WSe_2$ exfoliation. M.J. performed $SiO_2$ thermal oxidation and provided initial fabrication support. L.H. and A.J.M. wrote the paper. All authors have given approval to the final version of the manuscript.

## Competing interests

The authors declare no competing interests.
