## [Transparent Peer Review file · Nature Communications]

Low Resistance P-Type Contacts to Monolayer WSe₂ through Chlorinated Solvent Doping

Corresponding Author: Professor Andrew Mannix

Version 0:

Reviewer comments:

Reviewer #1

(Remarks to the Author)

In this work, the authors present a p-type doping strategy for monolayer WSe₂ transistors using chloroform solvent, demonstrating significant performance enhancements including high hole current (>200 $\mu\text{A}/\mu\text{m}$), reduced contact resistance (~2.5 $\text{k}\Omega\text{-}\mu\text{m}$), and maintained high current on/off ratio (1010) at room temperature. The study systematically investigates the physisorption mechanism and long-term stability (exceeding 8 months) through comprehensive characterization techniques (electrical measurements, optical spectroscopy, AFM, XPS) combined with DFT simulations. While this research provides valuable insights into 2D semiconductor polarity control, concerns regarding mechanistic clarity and application potential lead this reviewer to question the work's novelty relative to existing literature. Specifically, the manuscript's innovation appears limited given prior reports on WSe₂ p-type doping techniques [Nano Lett. 2023, 23, 10236–10242; ACS Nano 2023, 17, 19709–19723; Nano Lett. 2024, 24, 13528–13533; Nano Lett. 2025, 25, 3571–3578; ACS Nano 2025, 19, 10244–10254]. The following issues require attention:

1. Interfacial Mechanism Clarification:

1) While AFM/AES data suggest chloroform intercalation at the WSe₂/oxide interface, direct evidence linking intercalation to contact resistance reduction is lacking. Cross-sectional STEM-EDS or in situ TEM characterization would strengthen this claim.

2) Although DFT indicates physisorption, potential chemical interactions (e.g., Cl substitution at Se vacancies) remain unaddressed. XPS depth profiling or TOF-SIMS analysis could differentiate between physical adsorption and chemical bonding.

2. Computational Model Limitations:

1) The DFT analysis appears oversimplified, considering only two molecular orientations (Cl-facing/H-facing) while neglecting coverage effects and substrate interactions (e.g., SiO₂/WSe₂ interface).

2) Furthermore, the observed performance degradation at 200 °C annealing requires quantitative analysis through activation energy calculations or desorption kinetics modeling to elucidate thermal stability limitations.

3) The Bader charge transfer (0.0125 - 0.0349 excess electrons) in DFT calculations seems inconsistent with the substantial device improvements. A detailed discussion reconciling this discrepancy is needed, particularly regarding potential roles of defect passivation versus band structure modulation.

3. CMOS Compatibility:

1) As the doping mechanism operates at the WSe₂/oxide interface, its compatibility with standard CMOS processes remains questionable.

2) The authors should address integration challenges and propose strategies for selective area doping or hybrid n-p device architectures (such as WS₂, MoS₂ n-type devices).

4. Process Parameter Optimization:

Critical fabrication parameters (chloroform concentration, immersion temperature, ambient conditions) require explicit documentation. Concentration-dependent performance studies would establish dopant dosage effects and process robustness.

5. Environmental Stability Assessment:

The 8-month stability data necessitate clarification regarding storage conditions (inert vs. ambient environment). Additional analysis of humidity/oxygen impacts on doping efficacy would strengthen the stability claims.

6. Electrical Properties:

For the monolayer WSe₂ transistors using chloroform doping, the other electrical properties should be provided, such as transconductance, mobility, and Schottky barrier height.

Reviewer #2

(Remarks to the Author)

The manuscript 'Low Resistance P-Type Contacts to Monolayer WSe₂ through Chlorinated Solvent Doping' reports a strategy to reduce the hole contact resistance for high performance p-type FETs based on 2D TMDs - an important step towards developing scalable next generation CMOS. I have the following issues with the paper that should be considered by the authors:

1. Doping level - What was the original doping level of their WSe₂ samples, and how much enhancement was obtained after solvent doping?
2. Mobility - The authors suggest improvement in electron mobility to the increased drain current. Again, can they quantify the improvement in mobility as a result of the treatment?
3. Substrate - Is there a role of the substrate/dielectric surface on the solvent induced doping? Would the doping levels be different for HfO₂ and SiO₂ as they are known to have different surface dopants to TMDs
4. Contact Resistance - Could the authors provide a benchmark figure of their contact resistance compared to previous reports?
5. The authors need to provide spectroscopic evidence of doping through XPS, and correlate changes in binding energy to doping levels estimated from FET measurements.

Answers to the above comments will allow others in the community to reliably use the technique proposed and achieve high performance WSe₂ p-FETs.

Reviewer #3

(Remarks to the Author)

Here, Lauren Hoang et al. report a p-type doping method for 2D transistors, in which chloroform molecules are intercalated between monolayer WSe₂ channels and gate oxides in field-effect transistors, serving as dopants. The hole currents increase by 100 times to over 200 $\mu\text{A}/\mu\text{m}$, with an on/off ratio of 10^{10} after doping—comparable to the highest reported values for p-type 2D transistors.

The overall performance of the p-type 2D transistors is impressive, and the mechanism underlying the high doping efficiency is well explained. While there are several other studies on surface charge transfer doping of p-type WSe₂ transistors, reports achieving both high hole currents (greater than several $\mu\text{A}/\mu\text{m}$) and high on/off ratios are very limited (IEEE Electron Device Lett. 43, 319 (2022)). I believe researchers in the 2D transistor community will find this study of high interest. However, I have several concerns, as outlined below.

Main Comments

1. Versatility of the doping method

Can this doping method be applied to WSe₂ with arbitrary interfaces while still achieving high doping efficiency and stability? The authors suggest that intercalation of dopants at the substrate–2D channel interface is key to attaining both. However, different interfaces may affect the intercalation process and the resulting doping efficiency. For instance, intercalation may be more difficult in non-transferred, as-grown WSe₂ films on dielectric substrates or on hexagonal boron nitride (hBN), which are useful for large-area electronics and high-mobility quantum devices, respectively. It would be valuable if the authors could provide supporting experimental data.

2. Stability

Compared to other doping processes such as substitutional doping or charge transfer doping using inorganic interlayers, one of the main drawbacks of molecular doping is limited stability. Can the p-type doping be retained through further fabrication steps, such as wet processing during lithography for CMOS integration? Currently, the chloroform doping affects the entire channel globally, without spatial control, resulting in a strong suppression of electron conduction (as shown in Fig. 2b). It indicates the likely need to fabricate n-type and p-type channels separately by sequential fabrication processes. These limitations should be discussed when evaluating the scalability and practicality of this doping approach for CMOS applications.

Additionally, the limited thermal stability ($\sim 200\text{ }^\circ\text{C}$) raises concerns regarding device operation. Joule heating during current flow in the channel could lead to dopant desorption. It would be helpful if the authors could comment on the retention characteristics of the doping over time under operational conditions.

3. Origin of low contact resistance

The authors discuss the impact of chloroform doping in reducing contact resistance, but they do not provide Schottky barrier height (SBH) values for Pd/WSe₂ contacts before or after doping. It would strengthen the study to include temperature-dependent measurements or thermionic emission fits to extract SBH values and offer a more quantitative understanding of the band alignment at the contacts.

Other Comments

4. Related to the previous comment, does intercalation also occur beneath the metal-contacted regions of WSe₂? If so, would this directly affect the Fermi level at the contacts?

5. As mentioned above, the doping strongly suppresses electron conduction (Fig. 2b). Could the authors provide insight into the mechanism behind this phenomenon?

Reviewer #4

(Remarks to the Author)

Version 1:

Reviewer comments:

Reviewer #1

(Remarks to the Author)

Overall, this work demonstrates a convenient method for p-type doping of WSe₂ using chloroform and shows improvements in contact resistance. While the authors have addressed my previous concerns by providing additional explanations, data, and details, I remain unconvinced that the modest electrical performance advances presented here are sufficient for publication in Nature Communications. In particular, the recent article on “Lan et al. In International Electron Devices Meeting (IEDM) 1–4 (IEEE, 2023)” reported significantly higher performance in enhancement-mode PMOS, including a higher current of 727 $\mu\text{A}/\mu\text{m}$ and lower contact resistance of 0.875 $\text{k}\Omega\cdot\mu\text{m}$. In comparison, the values reported in this manuscript—2.5 $\text{k}\Omega\cdot\mu\text{m}$ and approximately 200 $\mu\text{A}/\mu\text{m}$ —though respectable, do not represent a substantial breakthrough. Furthermore, the practical applicability of this approach in CMOS integration remains uncertain. Therefore, I believe this study does not meet the high threshold for novelty and impact expected for Nature Communications.

Reviewer #2

(Remarks to the Author)

I appreciate the authors response to the concerns raised. I have the remaining issues with their responses:

1. Doping level: My concern regarding the initial doping level is motivated by the goal of extending this technique to WSe₂ obtained from alternative growth methods (MOCVD, ALD, exfoliation). Since process parameters for chloroform treatment may vary with the starting material, an estimate of the initial doping level in the present study would assist future researchers in adapting and reproducing the work. Without such context, reproducibility across different WSe₂ sources may be challenging. Could the authors address this issue?
2. The explanation regarding contact-limited behaviour is reasonable. However, since mobility values (even if underestimated) are routinely reported in WSe₂ FETs, could the authors provide an approximate pre-doping mobility range or at least benchmark their post-doping values against reported literature?
3. A major challenge with this work is the incompatibility of this technique with standard solvent based fabrication processes. The authors suggest encapsulation techniques might offer localised doping. However, their comment in Page 23 about chloroform intercalation through percolation underneath the contact interface challenges their previous assertion related to encapsulation, as lateral percolation would still occur. The authors should address this inconsistency.

Reviewer #3

(Remarks to the Author)

The authors have fully addressed my previous concerns. I now recommend the publication of this work in Nature Communications.

Reviewer #4

(Remarks to the Author)

We are grateful to the Reviewers for their time and careful evaluation. In this detailed Response Letter, we have carefully considered each Reviewer's comments and have revised our manuscript accordingly. The original Reviewer comments are in blue italics, followed by our responses in black text. Revisions to the manuscript text are shown in red text, below and in the resubmitted manuscript and supplement.

Reviewer #1 (Remarks to the Author):

In this work, the authors present a p-type doping strategy for monolayer WSe₂ transistors using chloroform solvent, demonstrating significant performance enhancements including high hole current (>200 $\mu\text{A}/\mu\text{m}$), reduced contact resistance ($\sim 2.5 \text{ k}\Omega\cdot\mu\text{m}$), and maintained high current on/off ratio (1010) at room temperature. The study systematically investigates the physisorption mechanism and long-term stability (exceeding 8 months) through comprehensive characterization techniques (electrical measurements, optical spectroscopy, AFM, XPS) combined with DFT simulations. While this research provides valuable insights into 2D semiconductor polarity control, concerns regarding mechanistic clarity and application potential lead this reviewer to question the work's novelty relative to existing literature. Specifically, the manuscript's innovation appears limited given prior reports on WSe₂ p-type doping techniques [Nano Lett. 2023, 23, 10236–10242; ACS Nano 2023, 17, 19709–19723; Nano Lett. 2024, 24, 13528–13533; Nano Lett. 2025, 25, 3571–3578; ACS Nano 2025, 19, 10244–10254]. [...]

We thank the Reviewer for their careful evaluation and constructive feedback, which has been extremely valuable in helping us to revise and enhance this manuscript. We agree that several prior studies have explored p-type doping of WSe₂, and we have now revised our manuscript to more clearly distinguish our work from these previous efforts. We respectfully submit that our method is distinct in its simplicity, reproducibility, and straightforward compatibility with monolayer device fabrication, and it achieves a rare combination of high performance and stability (both over long time scales and with thermal annealing). Additionally, we have provided mechanistic insights which sets this work apart from previous reports.

To directly address the other techniques:

- P. Ho et al. (*Nano Lett.* 2023) uses HAuCl₄ doping, which deposits metallic Au residue on the surface. This method is highly sensitive to the HAuCl₄ concentration and exposure time, leading to challenges in precise control and device variability.
- A. Oberoi et al. (*ACS Nano* 2023) demonstrates WO_x doping on bilayer WSe₂ with much lower drive currents ($I_D \sim 10 \mu\text{A}/\mu\text{m}$ in bilayer WSe₂ devices, compared with $I_D > 200 \mu\text{A}/\mu\text{m}$ in our monolayer WSe₂ devices). This work does not address the long-term dopant stability, but previous works on metal oxide doping (MoO_x^{1,2}, WO_x³) have been unstable over time.
- I. Kim et al. (*Nano Lett.* 2024) used PtCl₄ on exfoliated bilayer WSe₂. Similar to HAuCl₄, this approach leaves metallic Pt residue on the surface and shows high sensitivity to processing conditions.
- Y.T. Lin et al. (*Nano Lett.* 2025) uses lithographically defined photoresist exposure to dope the monolayer WSe₂. This adds fabrication complexity through another lithography step and introduces a polymer layer on the WSe₂ surface, which pose challenges for downstream CMOS fabrication.

¹ Xie, L. *et al.* Electrical measurement of non-destructively p-type doped graphene using molybdenum trioxide. *Appl. Phys. Lett.* 99, 012112 (2011).

² Cai, L. *et al.* Rapid flame synthesis of atomically thin MoO₃ down to monolayer thickness for effective hole doping of WSe₂. *Nano Letters* 17, 3854–3861 (2017).

³ Yamamoto, M., Nakaharai, S., Ueno, K. & Tsukagoshi, K. Self-Limiting Oxides on WSe₂ as Controlled Surface Acceptors and Low-Resistance Hole Contacts. *Nano Letters* 16, 2720–2727 (2016).

- K. Kanahashi et al. (*ACS Nano* 2025) uses substitutionally doped Nb-WSe₂ during growth, which is not amenable to post-growth tuning and requires more complex synthesis and fabrication protocols.

In contrast, our approach uses a single-step chloroform exposure that is rapid, scalable, and requires no additional lithography, metal deposition, or high-temperature processing. It yields monolayer WSe₂ transistors with hole current > 200 $\mu\text{A}/\mu\text{m}$, reduced contact resistance ($\sim 2.5 \text{ k}\Omega\cdot\mu\text{m}$), and an on/off ratio exceeding 10^{10} at room temperature. The doping effect remains stable for over 8 months without encapsulation. None of these other methods provide this set of useful properties.

To further clarify the doping mechanism and support the interpretation of physisorption-dominated behavior, we have added new x-ray photoelectron spectroscopy (XPS) measurements, updated density functional theory (DFT) simulations, new photoluminescence (PL) mapping and spectroscopy data, and additional electrical characterization (see revised **Supplementary Figures S6, S11–S13**). These results show no evidence of strong chemical bonding or substitution, consistent with a physisorption-based charge transfer mechanism.

We believe that the combination of robust performance, mechanistic insight, ambient stability, and fabrication simplicity provides a clear advance relative to the current literature.

[...] The following issues require attention:

1. Interfacial Mechanism Clarification:

1) While AFM/AES data suggest chloroform intercalation at the WSe₂/oxide interface, direct evidence linking intercalation to contact resistance reduction is lacking. Cross-sectional STEM-EDS or in situ TEM characterization would strengthen this claim.

We thank the Reviewer for raising the important question of interfacial mechanism and the role of intercalation in contact resistance reduction. We agree that direct nanoscale imaging methods, such as cross-sectional STEM-EDS or in situ TEM, could in principle help to visualize interfacial intercalation. However, in this system, both approaches present significant practical limitations. Focused ion beam (FIB) preparation of cross-sections can alter interfacial chemistry and drive desorption of volatile species such as chloroform, while in situ TEM requires prolonged electron-beam exposure under high vacuum, which can likewise induce desorption or decomposition of molecular adsorbates and obscure the native interfacial state. Due to these challenges, the in situ imaging of the interfacial mechanism is beyond the scope of this work.

Although these constraints limit the feasibility of direct structural imaging, our conclusions are supported by several consistent and complementary observations: AFM step height changes at the flake periphery after chloroform treatment are consistent with interfacial modification; AES depth profiling reveals changes in chemical composition near the buried interface, consistent with molecular species at that interface; and the reduced contact resistance remains stable for months under ambient conditions, indicating a persistent modification rather than transient surface adsorption.

These findings are in line with previous reports of molecular interfacial modification in van der Waals systems, such as graphene on SiO₂ (Kim et al., ref. 4). While we cannot claim to unambiguously attribute the effect solely to intercalation, the combined structural, chemical, and electrical evidence strongly supports a buried interface modification mechanism as the most plausible explanation for the observed

⁴ H. Kim et al., "Substrate-induced solvent intercalation for stable graphene doping." *ACS Nano* 7, 1155 (2013).

improvement in contact resistance. We have revised the manuscript to adopt a more conservative tone while retaining this interpretation as the most likely mechanism. On **page 1** in the abstract: “Auger electron spectroscopy and atomic force microscopy **suggest that** chloroform intercalates at the WSe₂ interface with the gate oxide, [...]”

2) Although DFT indicates physisorption, potential chemical interactions (e.g., Cl substitution at Se vacancies) remain unaddressed. XPS depth profiling or TOF-SIMS analysis could differentiate between physical adsorption and chemical bonding.

We thank the Reviewer for highlighting the possibility of chemical interactions, such as Cl substitution at Se vacancies. Addressing this suggestion has enabled us to strengthen the analysis and narrative of our manuscript. To investigate this possibility, we have **performed additional XPS measurements** on WSe₂ before and after chloroform exposure (**Figure R1.1, added as new Supplementary Fig. S11d-e**). We observe no new features or satellite features in the W 4f or Se 3d core level spectra following 15 hours of chloroform doping, aside from modest redshifts of 0.26 eV (W 4f) and 0.22 eV (Se 3d) consistent with *p*-type doping. These results suggest that chloroform treatment does not result in a significant change in the chemical bonding environment for the WSe₂ film, which is consistent with a physisorption-dominated mechanism rather than covalent functionalization or substitution.

Figure R1.1. **a**, XPS spectra of W 4f peaks on undoped and doped WSe₂ on SiO₂. The W 4f peaks redshift by 0.26 eV, consistent with *p*-type doping. **b**, XPS spectra of Se 3d peaks on undoped and doped WSe₂ on SiO₂. The Se 3d peaks redshift by 0.22 eV. For the Se 3d peak fit, fitting the spectra with 2 instead of 4 peaks led to the same shift value. No new features or satellite features in the W 4f or Se 3d peaks were observed after doping. This suggests that the chloroform doping did not significantly change the chemical bonding environment for WSe₂, consistent with physisorption-dominated mechanism rather than covalent functionalization or substitution.

We further note that low temperature PL measurements (**Fig. 3a,b** and **Supplementary Fig. 5c**) do not show spectral signatures associated with defect passivation^{5,6,7} (e.g., enhanced quantum yield, sharper excitonic peaks, or suppression of sub-bandgap emission). Instead, we observe quenched exciton intensity, enhanced trion features, and broader peak shapes. These spectral signatures are consistent with charge transfer via physisorption, not defect healing. We have clarified this further in the main text on **page 7**:

⁵ M. Amani et al. "Near-unity photoluminescence quantum yield in MoS₂." *Science* 350, 6264, 1065-1068 (2015).

⁶ A. Schwarz et al. "Thiol-based defect healing of WSe₂ and WS₂." *npj 2D Materials and Applications* 7, 1, 59, (2023).

⁷ J.H. Park et al. "Defect passivation of transition metal dichalcogenides via a charge transfer van der Waals interface." *Science advances* 3, 10, e1701661 (2017).

“The decrease in exciton intensity and emergence of bound states suggests that the passivation of WSe₂ defects is unlikely. Techniques⁵¹⁻⁵³ that passivate defects commonly exhibit an increase in quantum yield and the suppression of low-energy defect peaks at low temperatures.”

This interpretation is further supported by additional DFT simulations (**Figure R1.2**), which yield adsorption energies in the range of 300 – 500 meV and equilibrium separations of 2 – 4 Å between chloroform and WSe₂. Such values are characteristic of strong physisorption.⁸ In contrast, chemisorption normally involves binding energies ≥ 1 eV and bond lengths < 2 Å. Moreover, physisorption is a reversible process, and we experimentally demonstrate this reversibility by thermal desorption of chloroform from WSe₂ (**Figure 4d-f**). We have demonstrated these features in **Figure R1.2b**, which has been incorporated as **Figure 3d** in the main text. On **page 7** we have added: “In comparison, chemisorption is characterized by stronger binding energies (≥ 1 eV) and shorter distances (< 2 Å)⁵⁷. Thus, the adsorption energies and equilibrium distance from 2.5 – 3.2 Å (**Figure 3d**) resembles strong physisorption.”

Figure R1.2. a, Potential energy for physisorption and chemisorption, based on the Lennard-Jones potential model. Reproduced from ref. **Error! Bookmark not defined.** b, Potential energy vs. distance for an adsorbed chloroform to monolayer WSe₂ in Cl-facing and H-facing orientations. The adsorption energy ranges from 300 – 500 meV and the equilibrium distance from 2-4 Å, consistent with strong physisorption. Inset: zoomed-out view of the potential energy vs. distance.

2. Computational Model Limitations:

1) *The DFT analysis appears oversimplified, considering only two molecular orientations (Cl-facing/H-facing) while neglecting coverage effects and substrate interactions (e.g., SiO₂/WSe₂ interface).*

We thank the Reviewer for suggesting these opportunities to provide more comprehensive DFT analysis. To address this point, we have conducted additional simulations that consider the coverage, orientation, and substrate.

Coverage effects: We performed new DFT simulations using 2×2, 3×3, and 5×5 WSe₂ supercells with 1 chloroform molecule per supercell, corresponding to chloroform coverages of 1/2, 1/3, and 1/5 molecules per WSe₂ primitive cell (**Figure R1.3**, added as **Supplementary Fig. S6e**). In the Cl-facing orientation, the charge transfer efficiency slightly increases (from 0.955% to 1.43%) for higher chloroform density but slightly decreases (from 3.51% to 2.64%) in the H-facing orientation. For both orientations, the adsorption energy slightly increases in magnitude for higher chloroform density (-468 to -544 meV for Cl-facing, -351 to -427 meV for H-facing). In all coverage levels, the Cl-facing adsorption energy is larger than the H-

⁸ T. Nam et al. "Atomic layer deposition of a uniform thin film on two-dimensional transition metal dichalcogenides." *Journal of Vacuum Science & Technology A* 38, 3 (2020).

facing orientation but has lower charge transfer efficiency. Overall, changing the chloroform density only has a small effect on the adsorption energy and charge transfer.

We wish to note that preparing **Figure R1.2b** (now **Figure 3d** in the updated SI), along with preparing the new DFT figures in this manuscript, required running a large number of new DFT simulations. Thus, we improved the computational efficiency by changing the number of k -points; we also updated the functional correctional used to model van der Waals interactions. These changes slightly altered the previous DFT results, e.g., slightly changing the Bader charge transfer results of the original structures; however, these changes do not qualitatively change our results or our conclusions.

Figure R1.3. Charge transfer efficiency and adsorption energy with respect to chloroform coverage in both Cl-facing and H-facing orientations. A 2×2 , 3×3 , and 5×5 WSe₂ supercell were simulated with a chloroform molecule, corresponding to $\frac{1}{2}$, $\frac{1}{3}$, and $\frac{1}{5}$ chloroform molecules per WSe₂ primitive cell.

Orientational stability: We introduced small perturbations to the rotational orientation of the chloroform molecule ($\sim 5^\circ$), then re-relaxed the system. The chloroform molecule consistently returns to a near-original position and total energy (to within ~ 2 meV), suggesting that the H- and Cl-facing orientations are stable minima, rather than artifacts of constrained initial conditions. We have incorporated this into the methods section of the manuscript on **page 14**: “Other orientations of chloroform on WSe₂ are certainly possible, although a rigorous test for the most favorable orientation remains outside of the scope of this study. We confirmed that the Cl-facing orientation is stable compared to other similar random orientations by rotating the chloroform molecule by 5 degrees (i.e., tilting the C-H bond 5 degrees relative to the surface normal) and then re-relaxing the system. The chloroform molecule returned close to its original position and total energy (to within 2 meV), suggesting that this orientation is indeed favorable.”

Substrate effects: Lastly, we attempted to extend our simulations to consider interactions with substrates by modelling the system with (crystalline) SiO₂ and HfO₂. However, we found that these simulations yielded results that were likely non-physical due to the fact that the crystalline surface is likely an unrealistic structure, e.g., the chloroform molecule dissociating to passive dangling bonds. Unfortunately, modeling the full amorphous surface would greatly increase the computational burden and would likely require a great deal of time to optimize.

Therefore, rather incorporate such simulations directly into the present work, we have updated the DFT discussion of our manuscript to identify substrate interactions as an interaction that we do not consider here, and suggest is as a topic for future work on **page 13-14**: “We note that in this work, we consider only

interactions between the chloroform and WSe₂ without an insulator or substrate. Including the amorphous SiO₂ and HfO₂ would improve the accuracy of our simulations by capturing chloroform-substrate interactions; however, simulating this interface is computationally demanding, and we leave it as a promising research direction.”

These comprehensive additional simulations reinforce and validate the charge transfer mechanism and adsorption of the chloroform to the WSe₂ surface.

2) Furthermore, the observed performance degradation at 200 °C annealing requires quantitative analysis through activation energy calculations or desorption kinetics modeling to elucidate thermal stability limitations.

We thank the Reviewer for pointing out this opportunity to supplement our analysis with quantitative chloroform desorption kinetics. In response, we have conducted additional activation energy calculations to quantitatively characterize the adsorption-desorption mechanism of chloroform on WSe₂.

Figure R1.4a shows the calculated energy profiles as a function of the distance between the chloroform molecule and the WSe₂ surface for both Cl- and H-facing orientations. These results confirm that the chloroform is physisorbed on the surface (as opposed to chemisorbed), as indicated by the adsorption energy (< 1 eV) and equilibrium > 2 Å (as mentioned in the previous discussion). **Figure R1.4b** compares non-activated adsorption to activated adsorption⁹. Noticeably, our calculations show that there is no additional barrier to desorption, meaning that the activation energy for desorption is equal to the adsorption energy. We can use the inverted Arrhenius equation for desorption to determine the typical residence time of an adsorbed molecule (τ) and the rate constant for desorption (k_{des}) by:

$$\tau = 1/k_{des} = v^{-1} \exp\left(\frac{E_{des}}{k_B T}\right)$$

We note that the frequency prefactor of desorption (v) is not known for our system. However, if we assume that v is independent of temperature, we can estimate the relative desorption rate at different temperatures. At 200 °C (473 K) and $E_{des} = 0.348\text{eV}$, the desorption rate is approximately 2.6×10^9 times faster than at 296 K, which is consistent with the rapid desorption rate we observe experimentally. The more stable orientation is the Cl-facing orientation, given the more negative adsorption energy. For the H-facing orientation, the desorption rate at 473 K is approximately $850 \times$ faster than at 296 K. We have highlighted this in the main text on **page 9**: “**This is supported by activation energy calculations (Figure 3d) which show that there is no additional barrier to desorption, leading to much faster desorption at higher temperatures.**” We agree that future experimental work such as temperature programmed desorption would further quantify the desorption kinetics. Meanwhile, our current analysis establishes an understanding of chloroform thermal stability in a WSe₂ device context.

⁹A. Kokalj. “Corrosion inhibitors: physisorbed or chemisorbed?,” *Corrosion Science*, 196, 109939 (2022).

Figure R1.4.a, Potential energy vs distance for an adsorbed chloroform to monolayer WSe₂ in Cl-facing and H-facing orientations. **b**, Difference between non-activated and activated adsorption for the persistence of the adsorbed molecule on the surface. Reproduced from ref. 9.

3) The Bader charge transfer (0.0125 - 0.0349 excess electrons) in DFT calculations seems inconsistent with the substantial device improvements. A detailed discussion reconciling this discrepancy is needed, particularly regarding potential roles of defect passivation versus band structure modulation.

We appreciate the Reviewer's insightful suggestion to quantitatively connect the predicted charge transfer in DFT calculations to the experimentally observed electrical behavior. In our devices, we observe a median threshold voltage (V_T) shift of +1V after chloroform doping, which corresponds to an induced hole density of $\sim 10^{13}$ cm⁻².

Using the DFT-derived Bader charge transfer values (0.0351 to 0.0095 holes per chloroform molecule from updated DFT simulation parameters) and the WSe₂ unit cell area of 9.532×10^{-16} cm² (based on $a = 3.3176$ Å from ref. 10), this induced hole density corresponds to a chloroform molecular coverage of 2.9 to 10.5×10^{14} cm⁻², or 0.27 to 1.00 chloroforms per WSe₂ unit cell. We have clarified this point in the main text on **page 7**: “For a threshold voltage shift of 1V, the calculated Bader charge transfer of 0.0095 (0.0351) holes per chloroform molecule corresponds to a surface coverage of 1.0 (0.27) chloroform molecules per WSe₂ unit cell.”

In summary, while the per-molecule charge transfer is relatively modest, the cumulative effect of moderate surface coverage leads to a substantial increase in hole density. The observed enhancement in I_D is partly attributable to this doping, but I_D is further enhanced by the decreased R_C and increased mobility which

¹⁰ C. Klinkert et al. “2-D Materials for Ultrascaled Field-Effect Transistors: One Hundred Candidate under the Ab Initio Microscope.” *ACS Nano* 14,7,8605-8615 (2020).

coincide with the doping process. Furthermore, as mentioned above, we do not believe that there is any evidence of defeat passivation of WSe₂ defect states based on our low temperature photoluminescence and XPS measurements.

3. CMOS Compatibility:

1) As the doping mechanism operates at the WSe₂/oxide interface, its compatibility with standard CMOS processes remains questionable.

We thank the Reviewer for raising this point. We agree that the CMOS compatibility of this doping technique remains an open question. Our intention is not to claim full CMOS integration readiness, but rather to demonstrate this simple and reproducible method for achieving high-performance *p*-type doping in WSe₂.

A key advantage of our approach, compared to previous methods using metallic salts (e.g., HAuCl₄, PtCl₄), is that it avoids leaving metal residues on the 2D surface, thereby minimizing contamination and metal diffusion (which are well-known challenges for CMOS integration). While the detailed form of future gate-all-around architectures for 2D materials remains an active area of development, we believe that advances like ours, which enable robust doping under mild conditions, without structural modification or metal incorporation, offer a valuable foundation for future integration strategies.

We have revised the main text on **page 11** to clarify: “We further assess the substrate dependence of our doping strategy and its implications for CMOS compatibility. While chloroform doping demonstrates reproducible *p*-type doping without leaving metallic residues on the surface, the sensitivity to standard wet-processing solvents (e.g. acetone) poses integration challenges (**Supplementary Fig. S12a**).”

Finally, we note that the ease of implementation and ambient temporal and thermal stability of the doping effect make this technique broadly useful for exploring low temperature charge transport in WSe₂, with potential applicability well beyond CMOS-specific contexts.

*2) The authors should address integration challenges and propose strategies for selective area doping or hybrid *n-p* device architectures (such as WS₂, MoS₂ *n*-type devices).*

We appreciate the Reviewer’s suggestion to address potential integration strategies, including selective-area doping and the formation of hybrid *n-p* device architectures. In response, we have added **Figure R1.5** as **Supplementary Fig. S12b**, which outlines two conceptual approaches for selective-area chloroform doping:

1. Gate-First or Encapsulation/sacrificial layer:

In this approach, a protective (sacrificial or dielectric) layer is deposited to encapsulate selected regions of the TMD, preventing exposure to the dopant via top surface adsorption or edge intercalation. After doping, the protective layer can be selectively removed to expose clean, undoped regions, enabling spatially defined *p*- and *n*-type domains. This method could be compatible with stacked or gate-first process flows, although challenges remain in identifying materials that effectively inhibit doping and in developing etching processes that preserve monolayer integrity.

2. Doping Everywhere then Removing Dopant:

An alternative strategy is to uniformly dope the TMD before the lithography step to define the contacts. After the contacts are patterned and deposited, the channel doping can then be selectively removed through other solvents (e.g. acetone, IPA) or high temperature thermal desorption. This approach ensures that doping only occurs below the contact region and maintaining an undoped channel. The primary challenge here is ensuring that the dopant remains stable under subsequent processing conditions, particularly at the contacts.

Figure R1.5: Proposed process flows for selective-area doping with chloroform doping. Top: gate-first or encapsulation/sacrificial layer approach. Here, a sacrificial layer is first deposited, protecting the TMD from undesired doping. Doping is then selectively applied to un-protected regions. Bottom: doping then removal approach. Here, the dopant is first applied then selectively removed using high temperature processes or other solvents.

These strategies highlight potential paths toward selective-area doping and hybrid n - p device architectures, including integration with n -type materials such as MoS_2 or WS_2 . We have included this discussion in the revised supplementary materials to encourage further exploration of these integration schemes.

4. Process Parameter Optimization:

Critical fabrication parameters (chloroform concentration, immersion temperature, ambient conditions) require explicit documentation. Concentration-dependent performance studies would establish dopant dosage effects and process robustness.

We thank the Reviewer for highlighting this opportunity to clarify our process and enhance the accessibility of this technique to the research community. In response to this comment, we have added detailed information to the Methods section on **page 11**: “The WSe_2 sample was immersed in **as-purchased, undiluted** chloroform (SIGMA-Aldrich, No. 650498), in a watchglass-covered borosilicate beaker **at standard room temperature (20-24°C) and relative humidity (approx. 25%-65%) in a ventilated fume hood. The chloroform solvent was used without modification (as-purchased $\geq 99.9\%$ chloroform).** Unless indicated otherwise, the doping process occurred overnight (> 8 hours).”

Regarding concentration-dependent studies, we note that chloroform is immiscible with water, and dilution with polar organic solvents such as acetone or isopropanol introduces interpretive challenges. These co-solvents, along with many others, are known to interact electronically with TMD surfaces and could act as co-dopants, thereby confounding our understanding of the role of chloroform in isolation (e.g., acetone could n -dope WSe_2). For this reason, all doping experiments were performed using neat (undiluted) chloroform. We now clarify this in the main text on **page 11-12**: “**Samples were exposed only to pure chloroform, as dilution with other solvents (e.g. acetone, IPA) could introduce co-dopant effects, complicating the interpretation of concentration-dependent trends.**”

We agree that further systematic studies on dose-dependent behavior would strengthen mechanistic understanding and may enable further process optimization. However, such studies will require careful design to isolate chloroform-specific effects and are beyond the scope of the current work.

5. Environmental Stability Assessment:

The 8-month stability data necessitate clarification regarding storage conditions (inert vs. ambient environment). Additional analysis of humidity/oxygen impacts on doping efficacy would strengthen the stability claims.

We thank the Reviewer for this helpful comment regarding the conditions used in our long-term stability study. We have clarified the storage conditions in the Methods section on **page 13**: “**For time stability measurements, devices were stored at room temperature in a nitrogen dry box under continuous N₂ purge, with relative humidity maintained between ~5%.**”

We agree that environmental factors such as humidity and oxygen are important considerations for long-term device reliability. While a comprehensive evaluation of ambient exposure lies beyond the scope of this work, which focuses on the doping mechanism and device performance, we recognize its relevance for future studies, particularly in the context of scalable device integration and future research applications of the chloroform doping technique.

We also note that encapsulation is routinely used in practical devices to mitigate environmental effects. As such, while ambient sensitivity is a relevant consideration, we do not expect it to fundamentally limit the viability of this doping method in properly packaged systems.

We have updated the text to reflect this on **page 13**: “**At present, the impact of ambient humidity and oxygen on the long-term stability of chloroform doping remains an open question and warrants future investigation, particularly in the context of encapsulated or integrated device architectures.**”

6. Electrical Properties:

For the monolayer WSe₂ transistors using chloroform doping, the other electrical properties should be provided, such as transconductance, mobility, and Schottky barrier height.

We thank the Reviewer for their suggestion to provide additional electrical properties of the chloroform-doped monolayer WSe₂ transistors. In response, we have acquired new temperature-dependent transport data, and we have included new analyses of transconductance, estimated mobility, and Schottky barrier height (SBH) in our revised manuscript.

Supplementary Fig. S3a shows the peak transconductance (g_m) before and after doping as a function of channel length. Our initial devices are strongly contact-dominated, as evidence in **Figure 2c** where drain current I_D does not increase with decreasing channel length. This means a precise quantification of mobility before doping is challenging. Thus, we refrain from explicitly quantifying mobility improvement. We estimate the mobility is around 22.7 to 36.2 cm²/(V·s) after doping through the peak transconductance values. This has been added as **Supplementary Fig. S3f** and shown here as **Figure R1.6**.

We also **acquired additional temperature-dependent transport data** to extract Schottky barrier height (SBH) before and after doping (**Figure R1.7**). However, across 5 doped and undoped devices, they do not exhibit the expected increase in $I_{\text{thermionic}}$ current in the deep subthreshold region. We believe that this method carries significant uncertainties for several reasons: Our devices have significant overlap between the thermionic emission and tunneling regimes.¹¹ As the screening length λ approaches 0, the thermionic and tunneling slopes merge and there is no clear regime to fit.¹¹ In our devices, owing to the thin body and scaled high- κ HfO₂ dielectric, λ is small. We can see in the transfer curves (**Figure R1.7a,d**) that there is no “knee” where we can accurately locate a flat band voltage (V_{FB}).

¹¹ P. Wu, J. Appenzeller. “Toward CMOS like devices from two-dimensional channel materials.” *APL Materials* 7, 10, (2019).

Figure R1.6. Field-effect mobility $\mu = g_m L_{ch} / (W C_{ox} |V_{DS}|)$ estimated from the forward and backward V_{GS} sweeps for $L_{ch} = 1 \mu\text{m}$ devices, where C_{ox} is the back-gate oxide capacitance per unit area. The mobility is estimated from the long channel devices to minimize the impact of contact resistance.

The WSe_2 devices are also highly ambipolar (**Figure 2b**). Thus in the off-state, the current is a contribution of the electron and hole branch which can complicate the extraction by violating the assumption that the current is purely dominated by hole thermionic emission. Additionally, Schottky barrier extraction through the Arrhenius equation exhibits large uncertainties due to fitting a line to the logarithm of the temperature-dependent current.

For these reasons, direct extractions of the SBH are likely error-prone. Nevertheless, theoretical studies have calculated R_C as a function of SBH for different carrier densities, and we can use these previous results to estimate our SBH, as we have extracted R_C and carrier density as $\sim 2.5 \text{ k}\Omega \cdot \mu\text{m}$ and $\sim 10^{13} / \text{cm}^2$ after doping with chloroform respectively. From **Figure 2b** of ref. 12, this combination of R_C and carrier density is achieved at a barrier height of ~ 100 to 200 meV , suggesting that our barrier height is in this range.

We believe that rigorous quantum simulations, such as those used in ref.12, are a good next step for further understanding the electrical properties of metals contacting chloroform-doped WSe_2 . This task is non-trivial, as tunneling, the full dielectric and electrostatic environment must be considered. Furthermore, the tunneling distance from WSe_2 to the metal contact is not precisely known. Thus, rigorously quantifying the barrier height is beyond the scope of the present study.

To address the Reviewer’s comment, we added the following discussion on **page 5** of the main text: “**We estimate a Schottky barrier height (SBH) of $\sim 100 \text{ meV}$ after doping, consistent with previous theoretical calculations⁴². However, direct extraction of SBH from temperature-dependent data remains challenging due to the small screening length and large contribution of tunneling current across the narrow barrier. Rigorous simulations to extract SBH are an important future topic to understand the R_C improvement in these and other doped WSe_2 devices.**”

¹² P. Baikadi et al. "Quantum transport study of transition-metal dichalcogenide top-contacted geometries investigating the impact of nonuniform doping, dielectric environment, and image-force barrier lowering." *Physical Review Applied* 22, 6, 064058 (2024).

Figure R1.7. Schottky barrier height (SBH) extraction with temperature-dependent measurements for $L_{ch} = 1 \mu\text{m}$ WSe_2 FETs on $\sim 5 \text{ nm}$ HfO_2 . (a-c) undoped and (d-f) doped. (a,d) Temperature-dependent I_D - V_{GS} measured at $|V_{DS}| = 0.1 \text{ V}$. (b,e) Arrhenius plots for $V_{GS} = -3.0 \text{ V}$ to -1.0 V in steps of 0.2 V . (c, f) Extracted hole SBH = $\phi_{B,p}$. The flat band (V_{FB}) was difficult to determine, potentially leading to errors in $\phi_{B,p}$.

Reviewer #2 (Remarks to the Author):

The manuscript 'Low Resistance P-Type Contacts to Monolayer WSe₂ through Chlorinated Solvent Doping' reports a strategy to reduce the hole contact resistance for high performance p-type FETs based on 2D TMDs - an important step towards developing scalable next generation CMOS. I have the following issues with the paper that should be considered by the authors:

1. Doping level - What was the original doping level of their WSe₂ samples, and how much enhancement was obtained after solvent doping?

We thank the Reviewer for this helpful comment regarding the doping level and induced carrier concentration. To address this comment, we have added additional analysis and figures to the main text and supplementary information.

Precise quantification of the initial doping concentration in the WSe₂ is inherently difficult, as fabrication-induced effects (e.g., adsorbates, annealing, processing history, etc.) can alter the concentration. Techniques such as capacitance-voltage (C-V) measurements or Hall measurements are challenging to apply reliably in ultrathin 2D devices. However, based on the transfer characteristics (**Figure 2b**), the as-fabricated devices exhibit more pronounced *n*-type behavior, with the *n*-branch emerging near $V_{GS} = 0$ V. The induced hole density from doping is sufficient to compensate this initial electron density and shift the devices into *p*-type operation.

We estimate the induced carrier density from the observed threshold voltage shift (+1.0 V) after chloroform doping. The HfO₂ gate stack has an EOT of ~ 1.2 nm from MIMCAP measurements. Accounting for uncertainties arising from the van der Waals gap at the interface (typically, ~ 0.3 nm), we consider an EOT range of 1.2 - 2.5 nm (if a 0.3 nm vdw gap¹³ is added with $\epsilon_r \sim 1$, EOT = 2.4 nm). Using this range, the estimated doping-induced hole concentration is estimated to be between 0.9 and 1.8×10^{13} /cm², as shown in **Figure R2.1** and included as **Supplementary Fig. S3b**.

Figure R2.1. Constant-current threshold voltage ($V_{T,cc}$) at $I_{D,cc} = 10$ nA/ μ m before and after *p*-doping. Devices show a median V_T shift of +1.0 V, corresponding to $\Delta n = C_{ox} \Delta V_T / q = 0.9 - 1.8 \times 10^{13}$ cm⁻² carriers induced (with $C_{ox} = 1.38$ to 2.8 μ F/cm²).

¹³ Z. Sun, C. Chen, J. A. Robinson, Z. Chen and J. Appenzeller, "A mobility study of monolayer MoS₂ on low- κ /high- κ dielectrics," 2023 Device Research Conference (DRC), Santa Barbara, CA, USA, 2023, pp. 1-2, doi: 10.1109/DRC58590.2023.10258241.

We have updated the main text on **page 5** as follows: “All devices demonstrate a positive V_T shift (V_T extracted at a constant current $10 \text{ nA}/\mu\text{m}$)³⁶, with a median shift value of 1.0 V (from -2.6 V to -1.6 V), consistent with p -doping (**Figure 2d**). This V_T shift corresponds to roughly $0.9 - 1.8 \times 10^{13} / \text{cm}^2$ holes induced from this doping technique (**Supplementary Fig. S3b**).”

2. Mobility - The authors suggest improvement in electron mobility to the increased drain current. Again, can they quantify the improvement in mobility as a result of the treatment?

We thank the Reviewer for this helpful comment, and we believe they are referring to the improvement in the hole mobility following doping. **Supplementary Fig. S3a** shows an improvement in peak transconductance (g_m) after doping.

However, our initial devices are strongly contact-limited, as shown in **Figure 2c** where drain current I_D does not increase with channel length. This means that extracting an accurate pre-doping mobility values is difficult. For this reason, we refrain from quantifying a mobility improvement factor. However, using the peak transconductance (g_m) from $1 \mu\text{m}$ channel-length devices, we estimate the mobility is around 22.7 to $36.2 \text{ cm}^2/(\text{V}\cdot\text{s})$ after doping. This has been added as **Supplementary Fig. 3f** and shown here as **Figure R2.2**.

Figure R2.2. Field-effect mobility $\mu = g_m L_{ch} / (W C_{ox} |V_{DS}|)$ estimated from the forward and backward V_{GS} sweeps for $L_{ch} = 1 \mu\text{m}$ devices, where C_{ox} is the back-gate oxide capacitance per unit area. The mobility is estimated from the long channel devices to minimize the impact of contact resistance.

3. Substrate - Is there a role of the substrate/dielectric surface on the solvent induced doping? Would the doping levels be different for HfO2 and SiO2 as they are known to have different surface dopants to TMDs

We thank the Reviewer for this insightful comment regarding the possible role of the substrate in solvent-induced doping. To address this point, we fabricated additional monolayer WSe_2 devices using SiO_2 as the back-gate dielectric and measured the effects of chloroform doping on this substrate (**Figure R2.3** added as **new Supplementary Fig. S13**). Like WSe_2 devices on thin HfO_2 , the devices on SiO_2 see substantial improvement in their p -type performance after chloroform doping, including: positive V_T shift, $\sim 100\times$ increase in I_D , and a median induced hole density of $\sim 9.8 \times 10^{12} / \text{cm}^2$ estimated based on their V_T shift. We have updated the main text on **page 11**: “To further clarify the influence of the substrate, we extend our experiments to include WSe_2 devices on SiO_2 . These show comparable performance enhancement as those on HfO_2 ($\sim 100\times$ increase of I_D , positive V_T shift, and similar change in carrier concentration) (**Supplementary Fig. S13**). This suggests that the doping mechanism is robust across different amorphous dielectrics.”

Figure R2.3. Monolayer WSe₂ devices on 100 nm SiO₂ before and after chloroform doping. **a**, Cross-sectional schematic of WSe₂ transistor (top) and device fabrication procedure (bottom). Monolayer WSe₂ grown by chemical vapor deposition on sapphire was wet-transferred onto 100 nm SiO₂. Contact pads, channel regions, and fine contacts were defined using electron-beam lithography. Pd/Au contacts were used for fine contacts. **b**, Measured I_D vs. V_{GS} before and after doping at several channel lengths (L_{ch}) from 0.1 to 1 μm . **c**, L_{ch} -dependent statistical analysis before and after doping of maximum drain-current $I_{D,max}$ at $V_{GS} = -60 \text{ V}$. **d**, Threshold voltage ($V_{T,cc}$) at a constant current of 10 nA/ μm before and after doping. The median V_T shift (ΔV_T) is -45.6 V, corresponding to $\sim 9.8 \times 10^{12} \text{ cm}^{-2}$ carriers induced (calculated from $\Delta n = C_{ox} \cdot \Delta V_T / q$). This closely matches the hole carriers induced on HfO₂ back-gates (**Supplementary Fig. S3b**). **e**, Maximum transconductance (g_m) vs. L_{ch} before and after doping.

4. Contact Resistance - Could the authors provide a benchmark figure of their contact resistance compared to previous reports?

We appreciate the Reviewer's suggestion to include a benchmark comparison for contact resistance. We have now added this as **Supplementary Fig. S15** which is replicated below as **Figure R2.4**. This plot compares the maximal hole current versus the contact resistance for monolayer WSe₂ devices at $V_{DS} = -1 \text{ V}$, across various contact metals and doping strategies. Notably, chloroform doping achieves state-of-the-art performance at both room temperature and cryogenic temperatures.

Figure R2.4. Benchmarking maximum p -type current $I_{D,\max}$ vs. contact resistance R_C for monolayer WSe_2 at $V_{\text{DS}} = -1$ V, using various contact metals and doping strategies. Circles mark results with no intentional doping^{9–12}, squares denote oxide-based doping (MoO_x , WO_x , NO_x)^{3,13–17}, and triangles label halide-based doping¹⁸. Filled markers indicate room temperature values and unfilled markers indicate low temperature < 10 K values. Our results with chloroform doping (stars) achieve among the highest hole currents and lowest contact resistance to date for monolayer WSe_2 .

5. The authors need to provide spectroscopic evidence of doping through XPS, and correlate changes in binding energy to doping levels estimated from FET measurements.

We thank the Reviewer for this suggestion and have conducted additional XPS measurements to assess doping-induced changes in the W and Se core-level binding energies. We have added this data, shown in **Figure R2.5**, as **Supplementary Fig. S11d-e**. After doping, we observe a ~ 0.26 eV redshift in binding energy for both the Se 3d and W 4f peaks. This shift to lower binding energy is consistent with a decrease in work function and p -type doping. Additionally, no new spectral features are observed in either the W or Se spectra, which suggests that no new chemical bonds are formed (consistent with physisorption).

Figure R2.5. a, XPS spectra of W 4f peaks on undoped and doped WSe_2 on SiO_2 . The W 4f peaks redshift by 0.26 eV, consistent with p -type doping. **b**, XPS spectra of Se 3d peaks on undoped and doped WSe_2 on SiO_2 . The Se 3d peaks redshift by 0.22 eV. For the Se 3d peak fit, fitting the spectra with 2 instead of 4 peaks led to the same shift value. No new features or satellite features in the W 4f or Se 3d peaks were observed after doping. This suggests that the chloroform doping did not significantly change the chemical bonding environment for WSe_2 , consistent with physisorption-dominated mechanism rather than covalent functionalization or substitution.

Answers to the above comments will allow others in the community to reliably use the technique proposed and achieve high performance WSe₂ p-FETs.

We thank the Reviewer for this comment. We have carefully addressed each point to improve clarity and provide sufficient detail for reproducibility. We hope these additions will help the broader community adopt and apply this doping technique for high-performance WSe₂ p-FETs.

Reviewer #3 (Remarks to the Author):

Here, Lauren Hoang et al. report a p-type doping method for 2D transistors, in which chloroform molecules are intercalated between monolayer WSe₂ channels and gate oxides in field-effect transistors, serving as dopants. The hole currents increase by 100 times to over 200 $\mu\text{A}/\mu\text{m}$, with an on/off ratio of 10^{10} after doping—comparable to the highest reported values for p-type 2D transistors.

The overall performance of the p-type 2D transistors is impressive, and the mechanism underlying the high doping efficiency is well explained. While there are several other studies on surface charge transfer doping of p-type WSe₂ transistors, reports achieving both high hole currents (greater than several $\mu\text{A}/\mu\text{m}$) and high on/off ratios are very limited (IEEE Electron Device Lett. 43, 319 (2022)). I believe researchers in the 2D transistor community will find this study of high interest. However, I have several concerns, as outlined below.

We thank the Reviewer for their positive and thoughtful assessment of our work, and for recognizing its interest and relevance to the 2D transistor community.

Main Comments

1. Versatility of the doping method

Can this doping method be applied to WSe₂ with arbitrary interfaces while still achieving high doping efficiency and stability? The authors suggest that intercalation of dopants at the substrate–2D channel interface is key to attaining both. However, different interfaces may affect the intercalation process and the resulting doping efficiency. For instance, intercalation may be more difficult in non-transferred, as-grown WSe₂ films on dielectric substrates or on hexagonal boron nitride (hBN), which are useful for large-area electronics and high-mobility quantum devices, respectively. It would be valuable if the authors could provide supporting experimental data.

We thank the Reviewer for this important comment regarding the versatility of the chloroform doping method across different substrates. To address this, we fabricated additional monolayer WSe₂ devices using SiO₂ as the back-gate dielectric and evaluated their performance after chloroform doping (**Figure R3.1** added as **new Supplementary Fig. S13**). Similar to WSe₂ devices on thin HfO₂, the WSe₂ devices on SiO₂ exhibit substantial p-type enhancement after chloroform doping, with a positive V_T shift and $\sim 100\times$ increase in I_D . Based on the V_T shift, the induced carrier density after doping ($\sim 9.8 \times 10^{12}$ /cm² holes) is also comparable to the WSe₂ devices on HfO₂ (**new Supplementary Fig. S3b**, reproduced here as **Figure R3.2**). We have updated the main text on **page 11** to state: “To further clarify the influence of the substrate, we extend our experiments to include WSe₂ devices on SiO₂. These show comparable performance enhancement as those on HfO₂ ($\sim 100\times$ increase of I_D , positive V_T shift, and similar change in carrier concentration) (**Supplementary Fig. S13**). This suggests that the doping mechanism is robust across different amorphous dielectrics.” We acknowledge that the behavior on highly crystalline or low-surface-energy substrates (e.g., hBN) or in as-grown WSe₂ films may differ, as the Reviewer suggests. We consider this an important direction for future work.

Figure 3.1. Monolayer WSe₂ devices on 100 nm SiO₂ before and after chloroform doping. a, Cross-sectional schematic of WSe₂ transistor (top) and device fabrication procedure (bottom). Monolayer WSe₂ grown by chemical vapor deposition on sapphire was wet-transferred onto 100 nm SiO₂. Contact pads, channel regions, and fine contacts were defined using electron-beam lithography. Pd/Au contacts were used for fine contacts. b, Measured I_D vs. V_{GS} before and after doping at several channel lengths (L_{ch}) from 0.1 to 1 μm. c, L_{ch} -dependent statistical analysis before and after doping of maximum drain-current $I_{D,max}$ at $V_{GS} = -60$ V. d, Threshold voltage ($V_{T,cc}$) at a constant current of 10 nA/μm before and after doping. The median V_T shift (ΔV_T) is -45.6 V, corresponding to $\sim 9.8 \times 10^{12}$ cm⁻² carriers induced (calculated from $\Delta n = C_{ox} \cdot \Delta V_T / q$). This closely matches the hole carriers induced on HfO₂ back-gates (**Supplementary Fig. S3b**). e, Maximum transconductance (g_m) vs. L_{ch} before and after doping.

Figure R3.2. Constant-current threshold voltage ($V_{T,cc}$) at $I_{D,cc} = 10$ nA/μm before and after p -doping. Devices show a median V_T shift of +1.0 V, corresponding to $\Delta n = C_{ox} \Delta V_T / q = 0.9 - 1.8 \times 10^{13}$ cm⁻² carriers induced (with $C_{ox} = 1.38$ to 2.8 μF/cm²).

2. Stability

Compared to other doping processes such as substitutional doping or charge transfer doping using inorganic interlayers, one of the main drawbacks of molecular doping is limited stability. Can the p -type doping be retained through further fabrication steps, such as wet processing during lithography for CMOS integration? Currently, the chloroform doping affects the entire channel globally, without spatial control, resulting in a strong suppression of electron conduction (as shown in Fig. 2b). It indicates the likely need to fabricate n -type and p -type channels separately by sequential fabrication processes. These limitations should be discussed when evaluating the scalability and practicality of this doping approach for CMOS applications.

We thank the Reviewer for raising this important point regarding stability during subsequent processing. We agree that molecular doping approaches, such as this one, can present integration challenges, particularly with respect to compatibility with standard wet processing steps.

To explore this point, we have conducted additional experiments on WSe_2 devices (on HfO_2 substrates), exposing them to acetone for ~ 8 hours after chloroform doping (**Figure R3.3, new Supplementary Fig. S12a**). We observe that this acetone exposure largely reverses the p -doping effect, as evidenced by the reduced I_D and re-emergence of the n -type conduction. This result is consistent with a physisorption-based doping mechanism that is reversible. While this is advantageous for process tunability, this limits the compatibility with uncontrolled wet processing steps.

We note that encapsulation strategies could help mitigate this issue and enable more robust integration. Furthermore, selective-area doping approaches that we now discuss in the new **Supplementary Fig. S12** may allow spatially controlled n - and p -type channel formation in future process flows.

To address this comment, we have added to the main text on **page 11**: “We further assess the substrate dependence of our doping strategy and its implications for CMOS compatibility. While chloroform doping demonstrates reproducible p -type doping without leaving metallic residues on the surface, the sensitivity to standard wet-processing solvents (e.g. acetone) poses integration challenges (**Supplementary Fig. S12a**).”

Figure R3.3. Measured I_D vs. V_{GS} curves of a monolayer WSe_2 device after doping with chloroform (green) and then after long (~ 8 hour) exposure to acetone (red). Devices revert back to similar currents as initial device performance (~ 1 - $10 \mu\text{A}/\mu\text{m}$ p -type drain current) and the n -branch re-emerges.

Additionally, the limited thermal stability (~200 °C) raises concerns regarding device operation. Joule heating during current flow in the channel could lead to dopant desorption. It would be helpful if the authors could comment on the retention characteristics of the doping over time under operational conditions.

We thank the Reviewer for raising this important point regarding dopant desorption under device operation and potential self-heating effects. While our thermal stability data suggests dopant desorption around ~200°C, we believe that this concern can be mitigated in practical applications for two reasons:

1. Encapsulation: In practical application, 2D devices are expected to be encapsulated, which will suppress dopant desorption and improve thermal stability.
2. Self-heating effects under operational conditions: Although self-heating can be significant in direct-current (DC) operation,^{14,15} it is substantially reduced under typical high-frequency (e.g. GHz) or digital switching applications. In these cases, the device on-state duration is generally shorter than the thermal time constant of the 2D device (typically 30 to 300 ns)¹⁶, preventing the device from reaching high steady-state temperatures. Therefore, dopant desorption from Joule heating is unlikely to be a major issue under typical operating conditions.

We have clarified this in the main text on **page 9**: “While this thermal budget could be a concern for direct-current (DC) operation with significant self-heating^{66,67}, devices operating under high frequencies will heat up less. This is because the device switching speed is higher than the 2D device thermal time constant (typically from 30 to 300 ns⁶⁸). In any case, this 150°C thermal budget for chloroform stability enables compatibility with oxide encapsulation by atomic-layer deposition (ALD), which often occurs between 100 to 200°C. This may also further enhance the thermal stability of the *p*-type doping, enable fabrication of top-gated devices, and allow for concurrent application of other doping techniques, such as solid charge transfer layers (e.g., MoO_x, WO_x).”

3. Origin of low contact resistance

The authors discuss the impact of chloroform doping in reducing contact resistance, but they do not provide Schottky barrier height (SBH) values for Pd/WSe₂ contacts before or after doping. It would strengthen the study to include temperature-dependent measurements or thermionic emission fits to extract SBH values and offer a more quantitative understanding of the band alignment at the contacts.

We thank the Reviewer for this suggestion. We performed additional temperature-dependent transport measurements to extract Schottky barrier height (SBH) values before and after doping (**Figure R3.4**). However, in both doped and undoped devices (across five samples), we do not observe a well-defined thermionic emission regime suitable for reliable SBH extraction.

This is due to several factors. The transport behavior shows significant overlap between thermionic emission and tunneling,¹⁷ particularly given the thin body and scaled high-k HfO₂ dielectric, which results in a short screening length (λ).¹⁷ As λ approaches 0, the thermionic and tunneling components merge, preventing clear separation. Additionally, the transfer curves (**Figure R3.4a,d**) lack a distinct “knee”, making it difficult to accurately determine the flat-band voltage (V_{FB}).

¹⁴ E. Yalon et al. “Energy Dissipation in Monolayer MoS₂ Electronics.” *Nano Letters* 17, 7, 3429-3433 (2017).

¹⁵ M. Wang, E. Pop. “Monte Carlo Simulation of Electrical Transport with Joule Heating and Strain in Monolayer MoS₂ Devices.” *Nano Letters* 25, 17, 6841-6847 (2025)

¹⁶ S. Islam et al. “Role of Joule Heating on Current Saturation and Transient Behavior of Graphene Transistors.” *IEEE Electron Device Letters*, 34, 2, 166 (2013).

¹⁷ P. Wu, J. Appenzeller. “Toward CMOS like devices from two-dimensional channel materials.” *APL Materials* 7, 10, (2019).

Figure R3.4. Schottky barrier height (SBH) extraction with temperature-dependent measurements for $L_{\text{ch}} = 1 \mu\text{m}$ WSe₂ FETs on $\sim 5 \text{ nm}$ HfO₂. **(a-c)** undoped and **(d-f)** doped. **a,d**, Temperature-dependent I_D - V_{GS} measured at $|V_{DS}| = 0.1 \text{ V}$. **b,e** Arrhenius plots for $V_{GS} = -3.0 \text{ V}$ to -1.0 V in steps of 0.2 V . **c, f**, Extracted hole SBH = $\phi_{B,p}$. The flat band (V_{FB}) was difficult to determine, potentially leading to errors in $\phi_{B,p}$.

The ambipolar nature of WSe₂ (**Figure 2b**) further complicates extraction, as the off-state current contains both electron and hole contributions. Finally, the two-step Arrhenius fitting method, involving fitting a logarithmic term, introduces substantial uncertainty in this regime.

For these reasons, we find that direct extraction of the SBH via temperature-dependent measurements is unreliable in our devices. However, prior theoretical studies provide a useful reference: R_C as a function of SBH has been modeled for various carrier densities. From **Figure 2b** of ref. 18, based on our measured R_C ($\sim 2.5 \text{ k}\Omega \cdot \mu\text{m}$) and carrier density after doping ($\sim 10^{13} \text{ cm}^{-2}$), these models suggest an effective SBH in the range of ~ 100 to 200 meV .

We also note that rigorous quantum transport simulations, such as those used in ref.18, can account for tunneling, electrostatics, and the detailed device geometry. These will be essential for accurately determining SBH in these devices and will therefore be an important direction for future work.

To address these points, we added the following discussion on **page 5** of the main text: “**We estimate a low Schottky barrier height (SBH) of $\sim 100 \text{ meV}$ after doping, consistent with previous theoretical calculations⁴². However, direct extraction of SBH from temperature-dependent data remains challenging**

¹⁸P. Baikadi et al. "Quantum transport study of transition-metal dichalcogenide top-contacted geometries investigating the impact of nonuniform doping, dielectric environment, and image-force barrier lowering." *Physical Review Applied* 22, 6, 064058 (2024).

due to the small screening length and large contribution of tunneling current across the narrow barrier. Rigorous simulations to extract SBH are an important future topic to understand the R_C improvement in these and other doped WSe₂ devices.”

Other Comments

4. Related to the previous comment, does intercalation also occur beneath the metal-contacted regions of WSe₂? If so, would this directly affect the Fermi level at the contacts?

We thank the Reviewer for this insightful question. We suspect that intercalation likely occurs, at least partially, beneath the metal-contacted regions of WSe₂. Since the metal contacts ‘seal’ the WSe₂ from the top and across any edges where the metal film crosses over, any intercalation would need to proceed laterally from the channel side of the contact interface. However, directly probing intercalation beneath the metal is experimentally challenging: techniques such as photoluminescence or auger electron spectroscopy cannot access this deeply buried interface. One possible strategy for future work would be to selectively block intercalation along the WSe₂ edges, or to use contact designs that allow for controlled interfacial exposure. Understanding the extent of intercalation at the metal/TMD interface, and its impact on Fermi level alignment, remains an important open question that we identify as a promising direction for future study.

5. As mentioned above, the doping strongly suppresses electron conduction (Fig. 2b). Could the authors provide insight into the mechanism behind this phenomenon?

We thank the Reviewer for this question. We believe that the suppression of electron conduction arises from two combined effects: (1) The introduction of hole carriers shifts the threshold voltage positively, requiring higher gate voltages for electron accumulation; (2) Doping modifies the band alignment at the metal/semiconductor interfaces, which reduces the Schottky barrier height (SBH) for holes (improving hole injection) while increasing the SBH for electrons (thereby suppressing electron tunneling).

We have updated the main text on **page 5** to clarify: “Noticeably, the electron branch is strongly suppressed after doping (**Figure 2b**). This suppression likely arises from the positive V_T shift from increased hole concentration (thus requiring higher V_{GS} for electron injection) and from the increase in electron Schottky barrier height, which together hinder electron current.”

Reviewer #4 (Remarks to the Author):

We thank the Reviewer for their time and effort in carefully evaluating our manuscript, and we appreciate their contribution to the review process.

Response to Reviewer 2's Comments

We are grateful for the Reviewer's time and constructive comments.

The original Reviewer comments are in *blue italics*, followed by our responses in black text. Revisions to the manuscript text are shown in *red text* below.

Reviewer #2 (Remarks to the Author):

I appreciate the authors response to the concerns raised. I have the remaining issues with their responses:

1. Doping level: My concern regarding the initial doping level is motivated by the goal of extending this technique to WSe₂ obtained from alternative growth methods (MOCVD, ALD, exfoliation). Since process parameters for chloroform treatment may vary with the starting material, an estimate of the initial doping level in the present study would assist future researchers in adapting and reproducing the work. Without such context, reproducibility across different WSe₂ sources may be challenging. Could the authors address this issue?

We thank the Reviewer for their constructive comments, which have helped us clarify key aspects of our study. We recognize the importance of assessing reproducibility across different WSe₂ sources and have conducted additional analyses to address this point. While the precise initial carrier concentration may vary between WSe₂ sources, its influence on the final carrier density after doping is expected to be limited, as the doping process is primarily governed by adsorption and charge transfer rather than by the preexisting doping level. Consistent with this, the >100× increase in current after doping suggests that the induced carrier population substantially exceeds the initial carrier density.

To provide quantitative context, we use a simple analytical model (ref [1]), which derives a simple expression to estimate the threshold voltage V_T of an ultra-thin n -type transistor:

$$V_T = \frac{\phi_M}{q} - \frac{\chi_S}{q} - \frac{Q_{SS}}{C_{ox}} + \frac{Q_{sd}}{C_{ox}} \quad (1)$$

where ϕ_M is the metal work function, χ_S is the semiconductor electron affinity, q is the elementary charge, and Q_{SS} is the fixed interfacial charge density. Q_{sd} is the depletion charge density, whose magnitude is proportional to the doping concentration (and is positive if the semiconductor is p -doped and negative if n -doped). We define the total initial (i.e., before doping with chloroform) charge density $Q_{total} = Q_{sd} - Q_{SS}$, and rearrange the above equation to obtain:

$$Q_{total} = Q_{sd} - Q_{SS} = \left(V_T + \frac{\chi_S}{q} - \frac{\phi_M}{q} \right) C_{ox} \quad (2)$$

To estimate Q_{total} , we use the measured transfer characteristics of the initial devices as seen in **Figure 2b**. The as-fabricated devices exhibit ambipolar n -type behavior, with the n -branch emerging near $V_{GS} = 0$ V.

¹ Jackson, Tom. "Thinking MOSFETs." *IEEE Transactions on Electron Devices* (2025).

We take an n -type $V_T \approx 1$ V at $L_{\text{ch}} = 1$ μm from **Figure 2b** and $\chi_S = 3.9$ eV for monolayer WSe₂ (ref [2]). The work function of thin (8 nm) Pt films is $\phi_M \approx 5.5$ eV (ref [3]) though the small amount of Ti in the gate stack could lower this; thus, we estimate ϕ_M could be between 5.0 to 5.5 eV. With these values, and assuming an EOT of 2.4 nm (including the contribution from the van der Waals gap), we estimate that $Q_{\text{total}} \approx -1.4 \times 10^{-7}$ to -8.6×10^{-7} C·cm⁻², or approximately 9×10^{11} to 5×10^{12} charges per cm². Thus, upon chloroform doping, the induced hole density of $\sim 10^{13}$ cm⁻² (**Supplementary Fig. 3b**) compensates this initial electron density, dominating the final carrier concentration and shifting the device to p -type operation.

We have **updated the main text on page 5**: “This V_T shift corresponds to roughly $0.9 - 1.8 \times 10^{13}$ /cm² carriers induced from this doping technique (**Supplementary Fig. 3b**). **Precise quantification of the initial doping concentration in the WSe₂ is difficult as fabrication-induced effects (e.g. adsorbates, annealing, processing history, etc.) can alter the concentration. We estimate an initial electron concentration on the order of 10^{12} cm⁻² prior to doping (see **Supplementary Note 1**). Thus, upon chloroform doping, the induced hole density of $\sim 10^{13}$ cm⁻² can compensate this initial doping and dominate the final carrier concentration, shifting the device to p -type operation.**”

Additionally, we have **added the above discussion to Supplementary Note 1**.

2. The explanation regarding contact-limited behaviour is reasonable. However, since mobility values (even if underestimated) are routinely reported in WSe₂ FETs, could the authors provide an approximate pre-doping mobility range or at least benchmark their post-doping values against reported literature?

We thank the Reviewer for this helpful suggestion. To provide a quantitative reference, we have inserted **Figure R2** as the **updated Supplementary Fig. 3f** to provide an approximate range of field-effect mobility for the initial undoped samples. From the maximum transconductance, we extract a mobility of $0.75 - 1.2$ cm²/(V·s). As noted in our earlier response, these devices are strongly contact-limited, and the extracted values therefore underestimate the intrinsic channel mobility. Accordingly, we prefer not to report a mobility enhancement factor after doping, since the apparent improvement would primarily reflect reduced contact resistance rather than a true change in intrinsic transport properties.

² Liu, Wei, et al. "High-performance field-effect-transistors on monolayer-WSe₂." *ECS Transactions* 58.7 (2013): 281.

³ Lee, Woo-Jae, et al. "Atomic layer deposition of Pt thin films using dimethyl (N, N-dimethyl-3-butene-1-amine-N) platinum and O₂ reactant." *Chemistry of Materials* 31.14 (2019): 5056-5064.

Figure R2. Field-effect mobility $\mu = g_m L_{\text{ch}} / (WC_{\text{ox}} |V_{\text{DS}}|)$ estimated from the forward and backward V_{GS} sweeps for $L_{\text{ch}} = 1 \mu\text{m}$ devices before and after doping (F and B indicate the forward and backward sweep directions, with the ranges shown in the inset). C_{ox} denotes the back-gate oxide capacitance per unit area. The mobility is extracted from long channel devices to reduce the impact of contact resistance. The initial devices are strongly contact-limited, making accurate pre-doping mobility extraction difficult. The estimated post-doping mobility is approximately 22.7 to 36.2 $\text{cm}^2/(\text{V}\cdot\text{s})$ at $V_{\text{DS}} = -1 \text{ V}$.

3. A major challenge with this work is the incompatibility of this technique with standard solvent based fabrication processes. The authors suggest encapsulation techniques might offer localised doping. However, their comment in Page 23 about chloroform intercalation through percolation underneath the contact interface challenges their previous assertion related to encapsulation, as lateral percolation would still occur. The authors should address this inconsistency.

We thank the Reviewer for the opportunity to clarify this point. We propose that chloroform intercalation can occur over a finite lateral distance beneath the metal-contacted regions of WSe_2 . In devices where the WSe_2 edges are exposed (*case 1*, **Figure R3** left), chloroform molecules may access the WSe_2 /oxide interface through these exposed edges and subsequently percolate laterally beneath the channel, extending a limited distance under the metal contact.

In contrast, for fully encapsulated geometries (*case 2*), no WSe_2 edges are exposed, and we therefore expect minimal opportunity for the chloroform to reach the WSe_2 /oxide interface. A third geometry (*case 3*) represents the localized doping configuration proposed in **Supplementary Fig. 12**, where partial encapsulation by oxide or metal could restrict intercalation to specific exposed edge regions, enabling spatially controlled doping.

Further experimental work will be needed to validate these hypotheses and to develop optimized encapsulation strategies for localized and controllable doping. In particular, directly imaging the dynamics of chloroform intercalation over time remains an experimental challenge that we are actively pursuing but is beyond the scope of the present study. We have added **Figure R3 to Supplementary Fig. S12** to illustrate the dependence of chloroform doping on encapsulation geometry

Figure R3. Schematic illustrating possible chloroform intercalation pathways in WSe₂ devices with different geometries. The gray rectangle indicates the substrate, the atomic structure of a representative WSe₂ region is indicated, and blue regions indicate encapsulation by a metal or dielectric film covering the WSe₂. Chloroform intercalation pathways are indicated by green arrows. *Case 1* (left): chloroform could enter from the WSe₂ exposed edges and extend a finite distance beneath the encapsulated regions. *Case 2* (middle): In fully encapsulated geometries, no WSe₂ edges are exposed, and doping is expected to be blocked. *Case 3* (right): Partial encapsulation by oxide or metal may enable localized doping.